# Real-time and immediate effects of backward walking exercise on pain intensity and lumbopelvic movement control in individuals with chronic non-specific low back pain with lumbar flexion syndrome

**Ellen Chan**[○]**, Lok-Yi Chan**[○]**, Hung-Kit Fong**[○]**, Yiu-To Mak**[○]**, Patrick Wai-Hang Kwong**[ID][‡]**, Eliza Rui Sun**[‡]**, Clare C. W. Yu**[ID][‡]**, Sharon M. H. Tsang**[ID][*][○]

Department of Rehabilitation Sciences, Faculty of Health and Social Sciences, The Hong Kong Polytechnic University, Hong Kong

[○] These authors contributed equally to this work.
[‡] PWK, ERS and CCWY also contributed equally to this work.
[*] Sharon.Tsang@polyu.edu.hk

## Abstract

### Objective

Backward walking may promote the preferential recruitment of lumbar extensors to optimize flexed spinal posture adopted LBP flexion subgroup. This cross-sectional study investigated the backward-walking exercise on a) real-time muscle activation, and b) its immediate effect on back pain intensity, movement control and lumbopelvic muscle activation in individuals with chronic non-specific LBP characterized with lumbar flexion syndrome.

### Method

Thirty adults with chronic non-specific LBP with clinical manifestation of flexion syndrome received assessments of their movement control at static standing and during the five-minute forward walking test, conducted before and after a 15-minute treadmill walking training in forward or backward direction (as the immediate effect), while real-time adaptation of the lumbopelvic muscles during walking training was also evaluated. Comparisons of back pain intensity, ratio of the normalized electromyography (EMG) of the paired lumbopelvic muscles during walking, and performance of the lumbar movement control tests (LMC) using inferential statistics to analyze between- and within-subject effects differences for the a) real-time and b) immediate effects of the single session of walking training designated in a backward direction as compared to forward direction.

**Data availability statement:** All relevant data are within the paper and its Supporting information files.

**Funding:** The author(s) received no specific funding for this work.

**Competing interests:** The authors have declared that no competing interests exist.

## Results

Two-way repeated measures analysis of covariance (ANCOVA) was adopted to minimize the confounding effect from covariates identified (maximal tolerable gait speed for EMG amplitude analysis, and gender for pain intensity and performance in lumbar movement control (LMC) tests). Significant post-training improvement in pain intensity (p = 0.014) and overall performance of the LMC tests (p = 0.006) were found for those who received backward walking training. Significant overall between-group effect (p = 0.022–0.026) and time-and-group interaction effect (p = 0.004–0.022) of ipsilateral internal oblique (IO) to multifidus (MF) ratio were found in swing phases of both legs. For ipsilateral erector spinae (ES) to rectus abdominis (RA) ratio, significant time effect (p = 0.022), between-group differences (p = 0.031), and real-time reduction during forward walking in left swing phase, and significant between-group differences (p = 0.024), time-and-group interaction effect (p = 0.009), and real-time increase during backward walking in right swing phase were noted. Real-time increasing trend during backward walking and real time decreasing trend of ipsilateral MF to ES ratio during forward walking were observed in stance phase of both legs, with time-and-group interaction effect at 6th (p = 0.007) and 12th minute (p = 0.006) during walking training in left stance phase.

## Conclusions

Backward walking exercise emerges to benefit LBP patients with lumbar flexion syndrome by inducing real-time increase in back extensors and deep stabilizing muscle recruitment together with the immediate post-training improvement in pain intensity and LMC test performance. Further research with longer training duration and larger sample size are recommended to better understand if greater and more sustainable therapeutic effect can be achieved with the walking exercise in backward direction for this specified LBP subgroup.

## Introduction

Low back pain (LBP) is the most common musculoskeletal disorder with the prevalence estimated to be 577 million globally [1]. According to the Global Burden of Disease Study 2019, LBP remained the leading cause of years lived with disability amongst all the diseases being studied [2]. As high as 90% of LBP disorders were diagnosed as non-specific LBP (NSLBP) due to the lack of a known pathoanatomical cause [3,4]. With the intrinsic heterogeneity underlying the NSLBP condition, clinicians found it challenging when prescribing interventions that are precise and specific to the root cause. Therefore, various classification systems of NSLBP were introduced to facilitate the matching between the impairment and intervention, by classifying patients into different subgroups.

The movement impairment-based classification system proposed by Sahrmann and associates categorized patients with NSLBP into different syndromes according

to the behavior of symptoms, spinal alignment, sequence, and relative contribution of mobility between interacting spinal and pelvic regions displayed during direction-specific movement tests and posture adoption [5]. Symptom-provoking movements were repeated to explore for reduction in symptoms after correction of alignment as confirmation test [5]. This classification system has proven to have moderate inter-rater reliability (κ ≥ 0.65) with ≥75% of agreement between examiners [6,7]. The five movement-based impairment syndromes specified by the direction(s) of provocative movements are rotation-extension, extension, rotation, rotation-flexion and flexion. Results of a meta-analysis revealed that patients with chronic low back pain generally had a smaller lumbar lordosis angle compared to healthy control, which is a typical clinical presentation of patients with lumbar flexion syndrome [8]. Studies also found that flexed lumbar spine increased passive spinal loading, raised intradiscal pressure, and compromised spinal stability [9,10], which demands timely intervention to prevent further deterioration of the condition. This study hence places a major focus on this subgroup of patients with NSLBP. The lumbar flexion syndrome was more common in men and young individuals [5]. Patients with flexion syndrome experienced an increase in symptoms during lumbar flexion and they tended to acquire a relatively flexed posture [5,7]. The lack of movement dissociation between the lumbar spine and lower limbs during functional activities, e.g., walking and hip flexion in standing, indicates the decrease in flexion-specific movement control of the lumbar spine displayed in this specified subgroup. A study revealed that the NSLBP population had significantly poorer performance in Lumbar Motor Control (LMC) tests compared to the healthy controls (Cohen d = 1.18), while the history of LBP was found to be positively correlated to the severity of the motor control impairment [11]. Therefore, integrating lumbar movement control tests for evaluation of the motor control impairment would assist in the categorization of the NSLBP subgroup of patients, selection of treatment strategy and evaluation of the effectiveness.

Although there were ample studies investigating the muscle recruitment pattern of patients with NSLBP using surface electromyography (EMG), only a few of them were targeted at those with lumbar flexion syndrome. Dankaerts et al. reported that patients with lumbar flexion syndrome demonstrated a significantly lower activation of lumbar multifidus, transverse fibers of internal abdominal oblique and iliocostalis lumborum and thoracis when compared to healthy subjects and lumbar extension syndrome patients [12]. NSLBP patients with lumbar flexion syndrome presented with significantly higher abdominal muscle, including internal abdominal oblique, activity across various postures and functional tasks, including sitting and standing [13] and sit-to-stand tasks [14,15]. These impaired muscle recruitment patterns substantiated the clinical manifestation of this patient subgroup featured with relatively lower multifidi activation and heightened abdominal muscle activity. However, the trunk muscle activity of patients with lumbar flexion syndrome in walking was yet to be explored.

Besides the well-known effect on aerobic capacity and physical fitness, different walking exercises have been prescribed as a treatment modality for LBP for re-education of movement patterns and motor control [16]. Additional therapeutic benefits associated with backward walking over the natural forward walking for patients with LBP contributed by the difference in spinal and hip mobility. The favorable kinematics included an increase in lumbar mobility in the sagittal plane [17] and greater hip extension in conjunction with lumbar extension during walking [18]. The extended posture in backward walking could possibly reverse the tendency of flexed posture in patients with lumbar flexion syndrome. The increase in both the hip and lumbar extension mobility could potentially improve lumbopelvic rhythm through regulating the relative contributions among hip followed by pelvic tilting and lumbar spine for the overall sagittal plane motion required during walking [19]. Such optimized lumbopelvic alignment came with positive augmentation of the muscle recruitment of both the lumbar multifidus and erector spinae when one walks in backward direction [20]. As recommended by Sahrmann and associates, patients classified with lumbar flexion syndrome commonly had weak and/or long back extensor muscles [5]. This particular LBP subgroup may therefore benefit from practicing backward walking, which helps enhance the relative stiffness of lumbar extensors.

Despite the potential benefits of backward walking on LBP patients have been advocated, there was limited research exploring the effect of backward walking on movement control. Although significantly greater paraspinal muscle activities

were found with the real-time electromyographic analysis of walking backward under water [21], its immediate effect remains unknown. Besides, the direct comparisons of activation of muscles with differential roles in spinal movement and stability, for example, superficial erector spinae and deep lumbar multifidus, respectively [22] are yet to be investigated. Moreover, to the best of our knowledge, the potential and applicability of backward walking exercise on flexion syndrome subgroups of NSLBP was yet to be investigated.

This study aims to evaluate the real-time modification of muscle recruitment patterns during backward walking, and the immediate effect on lumbar movement control and muscle recruitment pattern after backward walking training in chronic NSLBP patients classified as lumbar flexion syndrome. We hypothesize that backward walking would induce significant enhancement in lumbar extensor muscles recruitment pattern and improvement in performance of the lumbar movement control tests in patients with lumbar flexion syndrome.

## Materials and methods

### Participants

Thirty-one participants were recruited from the local community with reference to the following inclusion criteria: aged 18–65, chronic NSLBP with/without lower limb radiation or leg pain for more than three months, able to walk independently without aids, presented with signs and symptoms of lumbar flexion syndrome (i.e., presented with the flexed or flat lumbar spine, the pain increases with lumbar flexion and/or rotation, and symptoms decrease with or are absent with movements or positions decrease in lumbar flexion) [5,11]. Participants with an inability to perform treadmill walking, true leg length discrepancy of more than 2 cm [23,24], history of neurological disorders which affect motor control (e.g., cerebrovascular accident), history of idiopathic scoliosis [25], history of major spinal/lower extremity surgery, and history of spinal/lower extremity fracture, were excluded. A total of 28 participants who fulfilled the inclusion criteria and clearance from any of the exclusion criteria would be required. The proposed sample size was calculated using G*Power 3.1.9.2 with the assumption of a medium effect size (f = 0.25) on the pre-to-post training changes in the Lumbar Motor Control test since there was no previous report on its effect size in walking training, with a level of significance set at 0.05 and power of 0.8. One of the recruited participants was unable to perform treadmill walking and thus was excluded. Thirty participants were included in the study. Participants were randomly allocated into two groups, namely the backward walking (BW) group and forward walking (FW) group.

This study was approved by the Institutional Review Board of The Hong Kong Polytechnic University. Prior to the commencement, a detailed explanation of the study procedures was given, followed by obtaining written consent from all participants. Data was anonymized using a coding system for analysis.

### Procedures

The experimental procedures were conducted by four registered physiotherapists at the Spine Research Laboratory of The Hong Kong Polytechnic University between 1 December 2022 and 25 April 2023. Demographic data and self-reported outcome measures were collected from all participants before the experimental procedures started. Measurements were done at three different time points: before the experimental condition, during the experimental condition with designated walking direction, and immediately after the experimental condition (Fig 1).

### Initial assessment

Participants were asked to rate their pain level using Numeric Pain Rating Scale 0–10 (NPRS 0–10) upon standard forward flexion movement. Participants were instructed to perform LMC tests, with the performance being video-recorded and later rated by two independent off-site assessors. Twelve pairs of EMG electrodes were applied to corresponding body landmarks after skin preparation. Prior to the initial forward walking assessment, a one-minute warm-up time was

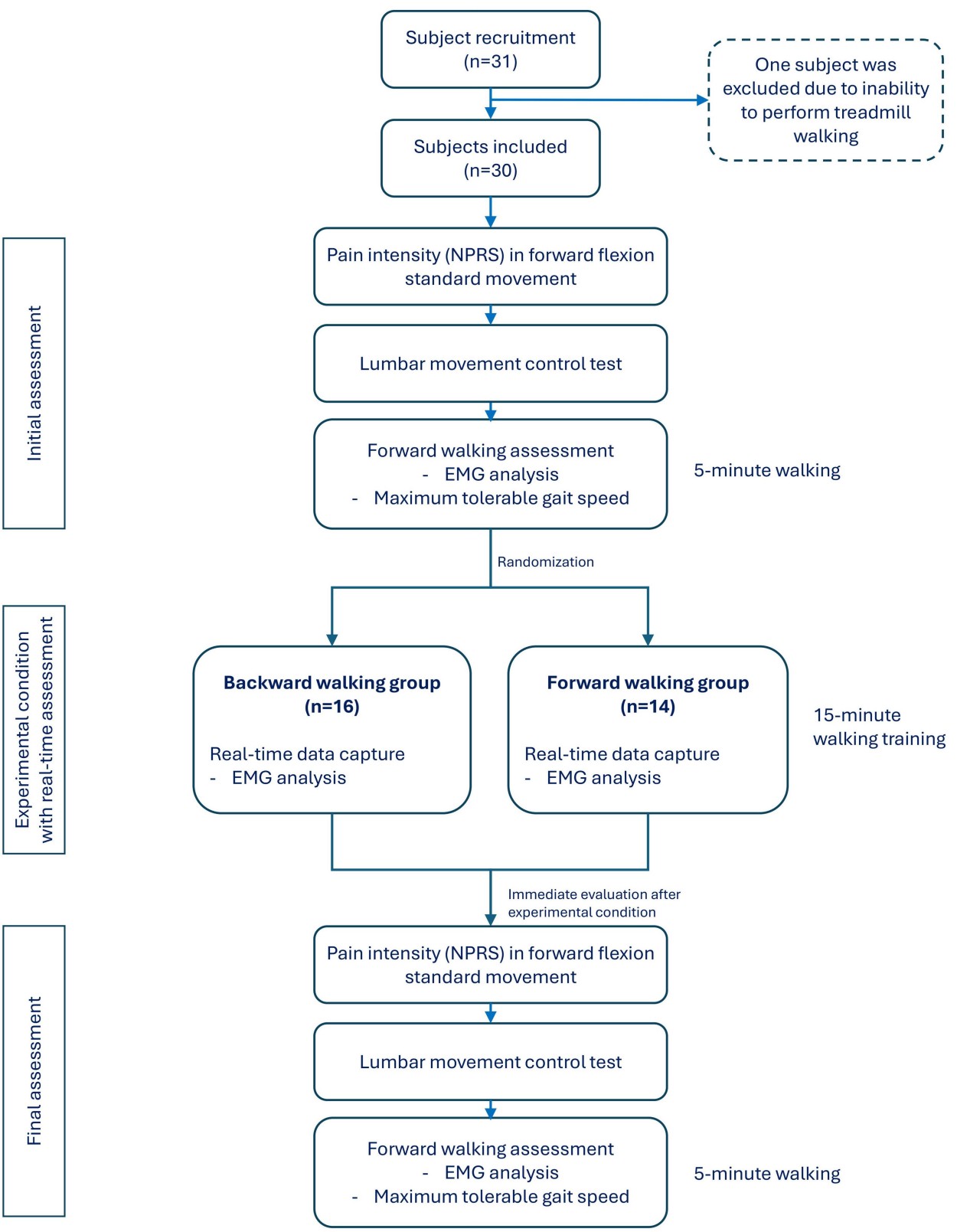

**Fig 1. Experimental procedures.**

granted. The speed of the treadmill gradually increased until participants reported that the speed was 'comfortable to walk without external assistance' and stopped increasing before the speed was 'too fast to walk'. Participants were blinded to the speed of the treadmill and were instructed to continue walking at the speed for five minutes. The maximal tolerable gait speed of participants was recorded for further analysis. The EMG signals of participants during initial walking assessment were recorded. Another one minute of walking was given as a cool-down period, with the treadmill speed being ramped down gradually.

**Experimental conditions.** Participants were randomly allocated to either the FW group or BW group by drawing ballots. After a ten-minute washout period, participants completed a 15-minute walking training, with an additional one-minute warm-up and cool-down period [17,26]. Participants allocated to the FW group performed walking in the forward direction, while those to the BW group walked backward.

In the BW group, participants were instructed to walk backward at their comfortable speed without external assistance. The treadmill speed gradually increased during the warm-up period until the preferred speed was reached, and then gradually decreased during the cool-down period. The procedures for the FW group were the same, except for the direction of walking. For safety, both experimental groups were closely monitored and supervised. A safety clip was attached to the participant's clothing, and a researcher stood by the treadmill to slow down or terminate the treadmill walking if necessary. EMG signals of the trunk muscles were recorded for one minute every three-minute intervals during the walking session after the warm-up period.

## Final assessment

The final assessment was performed immediately after the 15-minute walking of the allocated experimental conditions (i.e., walking in a forward or backward direction), using the same assessment procedures conducted at the initial assessment described above, and detailed in Fig 1 and the instrumentation and measurements section below.

## Instrumentation and measurements

**Demographics.** Demographics of the participants including age, gender, height, weight, body mass index (BMI), duration of LBP, and score of the set of three self-administered questionnaires, namely Roland-Morris Disability Questionnaire (RMDQ), Fear Avoidance Belief Questionnaire (FABQ), and physical activity level in the form of International Physical Activity Questionnaire – Short Form (IPAQ-SF) were collected. Pain intensity reported during forward flexion test was the primary outcome using the Numeric Pain Rating Scale (NPRS 0–10) while the performance of the lumbar movement control tests and the recruitment pattern of the lumbar pelvic muscles were analyzed as the secondary outcomes.

## Lumbar movement control tests

**Number of lumbar movement control tests rated as incorrect.** A battery of four motor control tests (standing lumbar flexion, sitting knee extension, chest drop and rocking backward) that are specific for flexion syndrome, and they have been validated to identify impairment of dynamic lumbopelvic control in chronic NSLBP. These tests have proven to have a moderate level of inter-rater reliability (κ ≥ 0.61) [27]. Details of the procedures, and values of sensitivity and specificity of the four motor control tests were presented in Supplementary I [11,28,29]. The test items are rated on a binary scale, i.e., each item is rated as "incorrect" if premature or excessive lumbar spine flexion is observed by assessors. The greater the number of LMC tests charted as incorrect, the poorer the lumbar motor control the subject acquires. Adelt et al. revealed that NSLBP patients with an average of one of four items rated as incorrect when performing the battery of LMC tests [28]. The number of LMC tests rated as incorrect for each participant was documented. Participants with less than or equal to one of the tests rated as incorrect were therefore excluded from the study. Test results were based on visual inspection of two independent off-site assessors who were blinded to the

allocated direction of the walking training of the participants. The performance of the participants was recorded in lateral view. The videotapes were independently reviewed by two off-site assessors for LMC score rating. Assessor training was conducted by providing the assessors with the standardized procedures of implementation of the LMC tests in which comprehensive list of criteria and rubrics for score rating of each of the four movement control tests prior to the data collection. Mock reviews were carried out with trial of the LMC tests on healthy individuals as well as video clips taken from symptomatic individuals using the standardized recording procedure. Each of the two independent assessors was required to review the video clips during the two-week training period to establish the inter-rater agreement between them. A third assessor was involved for the rating of the LMC score in the presence of disagreement between the two assessors for rating of the corresponding test performance.

**Electromyographic analysis of lumbopelvic muscles.** Myoelectric activity of six pairs of selected trunk and gluteal muscles were examined during the walking assessments and training using the surface EMG method (MyoMuscle System, Noraxon Inc., USA), at the sampling frequency of 1000 Hz. Rectangular-shaped bipolar Al/AgCl gel electrodes were applied to bilateral internal abdominal oblique (IO), external abdominal oblique (EO), rectus abdominis (RA), erector spinae (ES), multifidus (MF) and gluteus maximus (GMax), with the inter-electrode distance of 2 cm (Table 1) [30–38]. The standardized skin preparation for surface EMG measurement was adhered [39]. Maximal voluntary isometric contraction (MVIC) of each targeted muscle was performed with an EMG signal recorded for normalization of EMG data with reference to the standardized procedures (Table 1) [30–37]. Assessor gradually increased the resistance until maximum contraction was reached. Participants were instructed to hold MVIC for three seconds for each target muscle [31].

EMG signals were recorded at the second minute during the initial and final forward walking assessment for detecting the immediate effect of the 15-minute walking training. For real-time changes during walking training, EMG signals were also collected at four intervals specified as the third, sixth, ninth and twelfth minute of walking training for evaluating the onset and course of the effect (Fig 2). The raw EMG signals were filtered with the bandpass range of 30–500 Hz [37]. Data were full-wave rectified and smoothed with the root-mean-square (RMS) value of the EMG signal 50ms [38]. RMS values of the signal amplitude of each target muscle group were normalized to the RMS value of recorded MVIC (i.e.,

**Table 1. Electrode placement of surface EMG and procedures of the maximal voluntary isometric contraction of the corresponding muscles.**

| Target muscles | Placement of EMG electrodes | Procedure of maximal voluntary isometric contraction (MVIC) |
| --- | --- | --- |
| Internal abdominal oblique (IO) | The lateral electrode is placed 1 cm medial to the anterior-superior-iliac-spine (ASIS). The other electrode is inferior to the horizontal line connecting ASISs with the lateral electrode medial to the inguinal ligament. [30,32,33,35] | Perform an oblique sit-up in crook-lying position attempting to move the resisted shoulder towards the opposite knee. (Testing ipsilateral IO and contralateral EO) [30,32,33,35] |
| External abdominal oblique (EO) | The upper border of the electrode is placed right below the most inferior point of the costal margin. The other electrode falls on the line joining the most inferior point of the costal margin and contralateral pubic tubercle. [30,32,33,35] | |
| Rectus abdominis (RA) | Vertical. 4 cm lateral to the navel. The lower border of the caudal electrode at the level of the navel. [30,32,33,35] | Perform partial sit-up in crook-lying position. Apply manual resistance to the shoulder towards trunk extension. [30,32,33,35] |
| Multifidus (MF) | 2 cm lateral to the spinous process of L5. Align with the line connecting PSIS and interspace between L1 and L2. [30,32,33,35] | Perform trunk extension in prone position. Apply manual resistance to the upper thoracic area. [30,32,33,35] |
| Erector spinae (ES) | Vertical. Two finger width (approximately 3 cm) lateral to the spinous process of L2. [30,32,33,35] | |
| Gluteus maximus (GMax) | Midpoint along the line connecting the second sacral vertebra to the greater trochanter. [36,37] | Perform resisted maximum-effort hip extension, performed with the subject lying prone on a treatment table, with the knee flexed 90°. [36,37] |

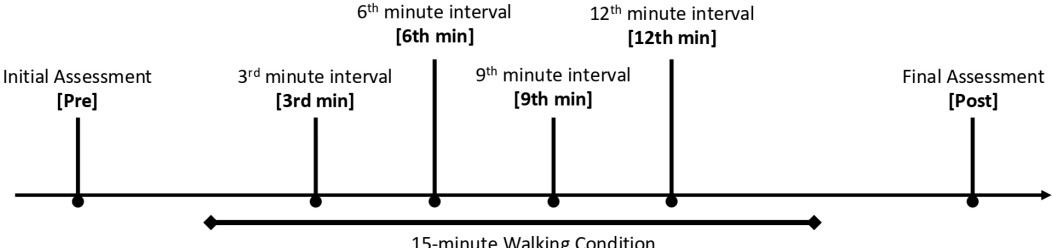

*Pre* - EMG signal during forward walking assessment
*3rd min, 6th min, 9th min & 12th min* - EMG signal during backward or forward walking training
*Post* - EMG signal during forward walking assessment

**Fig 2. Specified intervals of EMG and spinal alignment measurements during the 15-minute walking training.**

percentage of MVIC). The EMG amplitude of all muscles at the stance phase and swing phase of each leg of the gait cycle was defined by the rhythmic activities of the gluteus maximus using the customized MATLAB code (MathWorks, Natick, MA) [40–42]. EMG amplitude of all muscle groups during the respective phase of the gait cycle was averaged for further analysis.

The ratio of the normalized EMG amplitude (i.e., in % MVIC) of six pairs of muscle groups was calculated to evaluate the relative contribution of involved muscles during different phases of gait. The muscle groups were selected for comparison according to the muscle action and functional role in the stability and motor control of the lumbopelvic region, which were lumbar flexor-lumbar extensor, deep stabilizing muscle-superficial muscle and lumbar extensor-hip extensor (Table 2).

**Maximum tolerable gait speed in forward walking.** The maximum tolerable walking speed was measured as a surrogate indicator to reflect participants' coordination during gait and functional performance. It was recorded during the initial and final forward walking assessment. Participants were instructed to walk at the fastest speed without external assistance while maintaining their walking as naturally as possible for five minutes. The speed of the treadmill gradually increased and was recorded once it reached the participant's maximum.

## Data analysis

Statistical analysis was conducted by SPSS statistical software version 26.0 (SPSS Software, Chicago, IL, USA). Data were presented as mean ± standard deviation (SD).

**Reliability test.** The performance of LMC tests of each participant were examined by two different assessors. Intraclass correlation coefficient [$ICC_{(2,2)}$] with confidence intervals of 95% was used to analyze the inter-rater reliability of

**Table 2. Muscle pairings for EMG amplitude ratio analysis at the stance and swing phase of the gait cycle.**

| Category | Muscle Pairings |
|---|---|
| Lumbar Flexor: Lumbar Extensor (deep and superficial layers) [43] | Ipsilateral Internal abdominal oblique (IO)/Ipsilateral multifidus (MF) |
| | Ipsilateral erector spinae (ES)/Ipsilateral rectus abdominus (RA) |
| Deep Stabilizing Muscle: Superficial Muscle (ventral and dorsal aspects of the lumbar spine) [44] | Ipsilateral Internal abdominal oblique (IO)/Ipsilateral rectus abdominus (RA) |
| | Ipsilateral Internal abdominal oblique (IO)/Ipsilateral external oblique (EO) |
| | Ipsilateral multifidus (MF)/Ipsilateral erector spinae (ES) |
| Lumbar Extensor: Hip Extensor (posterior longitudinal sling and posterior oblique sling) [45] | Ipsilateral erector spinae (ES)/Ipsilateral gluteus maximus (GMax) |

the LMC test scores between assessors. An alpha level of 0.05 was adopted. An ICC greater than 0.75 indicates good reliability.

Two sets of data were captured at each time point of assessment of EMG and spinal alignment to establish corresponding intra-rater reliability. Intraclass correlation coefficient [$ICC_{(3,2)}$] with confidence intervals of 95% were adopted. An alpha level of 0.05 was adopted. An ICC value greater than 0.75 indicates good reliability.

**Number of LMC tests rated as incorrect.** Two-way repeated measures analysis of variance (ANOVA) was used to compare the number of LMC tests rated as incorrect between the two groups before and after the walking training, and the time-and-group interaction effect. Assumptions for repeated measures ANOVA were checked: 1) Normal distribution of each dependent variables was checked by Kolmogrov-Smirnov test; 2) Sphericity assumption for within-subject measurement to be checked by Mauchly's Test, and Greenhouse Geisser epsilon adjustment was used if the Sphericity test is violated; 3) Homogeneity assumption of dependent variables across group was checked by Levene's Test of Equality of Error Variances. The data failed to fulfill the above assumptions. However, since there is no non-parametric analog of two-way repeated measures ANOVA, it was still adopted for analysis of the number of LMC tests rated as incorrect and maximum tolerable gait speed despite known risk of statistical error. Alpha level is set at 0.05. Post-hoc independent t-test was used to analyze the between-subject effects if significance was identified. Contrast analysis was adopted to analyze the within-subject effects. As measurements were done at two intervals, the level of significance was adjusted to 0.025 by Bonferroni correction.

**Improvement in performance of the LMC tests.** The number of participants with and without improvement in performance of the four LMC tests were calculated and analyzed. Generalized Linear Mixed Model (fixed effect: direction of walking condition), with a binomial distribution and logit link, was used to evaluate the effects of walking direction on the changes in individual LMC tests (i.e., improved or not improved after the forward or backward walking condition).

**Normalized EMG data during walking.** Two-way repeated measures ANOVA were adopted to compare normalized EMG data between the two groups at six specific intervals, including initial assessment, final assessment, and the third, sixth, ninth and twelfth minute of walking training. The assumptions and procedures of analysis were the same as the above-mentioned. For post-hoc independent t-test and contrast analysis, the level of significance was adjusted to 0.0083 for measurement at six time points.

**Adjustment for potential confounding factors.** Chiu and Wang (2006) also proved that changes in gait speed and gender significantly affect muscle activity during normal walking [46]. Previous study also revealed that gender significantly affected the prevalence of lumbopelvic movement control impairment, with a greater proportion of male subjects showing impaired movement control [47]. Thus, a two-way repeated measures analysis of covariance (ANCOVA) was adopted to eliminate the effect of gait speed and gender, which are possible covariates to the outcome measures. The results reported in the subsequent section are based on findings in the corresponding analysis if adjustment for covariates is indicated.

## Results

Table 3 shows the demographics and clinical characteristics of the recruited participants, no significant baseline differences were found between the two walking training groups. Among the thirty participants included in the study, 20 (66.67%) were male and 10 (33.33%) were female The mean age was 28.67±5.91 years (ranging from 20 to 51 years). For the location of pain, fifteen (50%) participants had central back pain. All of them reported low back pain without any referred symptoms at their buttocks, thighs, or legs. Most of the participants reported low functional limitation or fear-avoidance belief, in which twenty-four (80%) of them scored <4 in RMDQ and reached the flooring effect [49]. Only four (13.33%) and one (3.33%) of them were classified as the subgroup of high fear avoidance belief in FABQ physical activity and work subscales, respectively. Eighteen (60%) participants reported high physical activity in IPAQ-SF. All participants were able to complete the 15-minute walking training in either forward or backward direction without any reported adverse response or aggravation of symptoms.

**Table 3. Demographics with mean (SD) except for gender, location of pain, disability (RMDQ), fear avoidance belief (FABQ) and physical activity level (IPAQ-SF).**

| Variables | | | BW Group (n=16) | FW Group (n=14) | p-value |
|---|---|---|---|---|---|
| | | | Mean (SD) | Mean (SD) | |
| Age (years) | | | 28.13 (4.978) | 29.29 (6.966) | .60 |
| Gender | Male | | 13 (81.25%) | 7 (50%) | .07 |
| | Female | | 3 (18.75%) | 7 (50%) | |
| Height (m) | | | 1.72 (0.101) | 1.68 (0.097) | .24 |
| Weight (kg) | | | 68.00 (8.268) | 63.61 (11.78) | .24 |
| BMI (kg/m²) | | | 22.98 (1.69) | 22.55 (3.00) | .63 |
| Pain | Intensity (NPRS) | | 3.69 (1.20) | 2.86 (1.70) | .13 |
| | Location of pain at lumbar region | Central | 8 (50%) | 7 (50%) | .96 |
| | | Left | 4 (25%) | 4 (28.57%) | |
| | | Right | 4 (25%) | 3 (21.43%) | |
| | History of low back pain (months) | | 12.13 (5.365) | 12.43 (7.449) | .90 |
| Disability | RMDQ | Total score | 2.25 (1.438) | 2.93 (3.43) | .58 |
| | | Flooring effect (score <4) | 14 (87.5%) | 10 (71.43%) | .27 |
| | | Ceiling effect (score > 20) | 0 (0%) | 0 (0%) | |
| Fear avoidance belief | FABQ-PA | Total score | 8.94 (4.89) | 10.00 (4.30) | .54 |
| | | Low fear avoidance belief (score ≤ 15) | 14 (87.5%) | 12 (85.71%) | .89 |
| | | High fear avoidance (score > 15) [60] | 2 (12.5%) | 2 (14.29%) | |
| | FABQ-W | Total score | 13.50 (8.25) | 16.21 (9.35) | .41 |
| | | Low fear avoidance belief (score ≤ 34) | 16 (100%) | 13 (92.86%) | .28 |
| | | High fear avoidance belief (score > 34) [61] | 0 (0%) | 1 (7.14%) | |
| Physical activities | IPAQ-SF | MET-min/week | 3419.63 (2304.35) | 2978.50 (2149.50) | .59 |
| | | Low activity | 0 (0%) | 2 (14.29%) | .29 |
| | | Moderate activity | 6 (37.5%) | 4 (28.57%) | |
| | | High activity | 10 (62.5%) | 8 (57.14%) | |

## Gait speed

For the spatiotemporal parameter, the BW group had a significantly higher baseline maximum tolerable gait speed in forward walking than the FW group did (p=0.02) (Table 4). The mean±SD of the gait speed of the training session was 2.08±0.54 km/h for backward walking group and 3.44±0.73 km/h for forward walking group. The gait speed during walking training in the BW group was significantly lower than the FW group (p<0.001).

## Reliability

**Lumbar motor control tests.** The ICC(2,2) value of the overall performance of LMC tests rated by two assessors was determined to be 0.827, indicating a good inter-rater reliability. The ICC of individual LMC tests, i.e., standing lumbar flexion, chest drop, sitting knee extension and rocking backward tests were 0.839, 0.565, 0.823 and 0.852, respectively. Only the chest drop test showed poor to moderate inter-rater reliability, while the other three tests all acquired good inter-rater reliability.

**Normalized EMG data.** The reliability of the EMG amplitude of all six pairs of muscles normalized in the percentage of MVIC was analyzed. The $ICC_{(3,2)}$ values range between 0.558 and 0.973 (Table 5). The result indicated that all EMG measurements demonstrated good to excellent repeatability except right ES, and hence, interpretation of findings related to analysis of the right ES requires adequate caution for its observed variability of EMG magnitude.

**Table 4. Maximum Walking Speed (km/h) during forward walking tests conducted at a) before and b) after the 15-minute walking training in allocated direction.**

| | BW Group | FW Group | Time Effect | Between-group Difference | | Time-and-group Interaction |
|---|---|---|---|---|---|---|
| | Mean (SD) | Mean (SD) | p-value (Observed Power) | p-value (Observed Power) | | p-value (Observed Power) |
| Pre | 5.73 (1.04) | 4.85 (0.90) | .62 (.08) | .03 (.60) | .02 | .27 (.20) |
| Post | 5.67 (0.89) | 5.02 (0.94) | | | .62 | |

**Table 5. The ICC values of normalized EMG amplitude of all six pairs of muscles.**

| | Internal oblique | External oblique | Rectus abdominis | Multifidus | Erector Spinae | Gluteus maximus |
|---|---|---|---|---|---|---|
| Left | 0.897 | 0.949 | 0.973 | 0.864 | 0.824 | 0.948 |
| Right | 0.938 | 0.926 | 0.971 | 0.833 | 0.558 | 0.922 |

## Covariances

**Maximum tolerable gait speed.** Differences in statistical significance of the ratio of the EMG amplitude of the paired muscles were identified after adjustment for maximum tolerable gait speed. Therefore, results of these listed outcome measures are presented based on ANCOVA with centered maximum tolerable gait speed as a covariate.

**Gender.** Although there was no statistically significant difference in gender between groups, the ratio of male to female (81.25% to 18.75%) was considered to be uneven in the BW group. Differences in statistical significance of pain intensity and performance in the LMC test were identified after adjustment for gender. Therefore, two-way repeated measures ANCOVA were used for further analysis with gender as a covariate for the listed outcome measures.

## Pain intensity in forward flexion standard movement

Significant overall time effect was found (p<0.001). Only the BW group showed a significant reduction in NPRS after walking training (p=0.014) (Table 6). While there was no time-intervention interaction effect, a nearly significant between-group effect was noted (p=0.06).

## Lumbar movement control tests

A significant time effect was found for the overall performance of the LMC tests (p<0.001). A significant improvement, i.e., reduction in number of LMC tests rated as incorrect after walking training, was shown only in the BW group (p=0.006). No between-intervention difference nor time-intervention interaction effect was found. For changes in individual LMC tests, only the sitting knee extension test demonstrated a significant difference (p<0.001), in which all subjects showed improvement was from the BW group (Table 7). Table 7 shows the results of the comparisons between two groups regarding the dichotomized effects (i.e., improved versus not improved) of the 15-minute walking training in specified direction on the LMC performance. Significantly greater number of participants reported to have improvement with their sitting knee extension performance for those who have received the backward walking training as compared to those received walking training in forward direction (p<0.001).

## Electromyography (EMG)

**Ratio of EMG signal amplitude.** Tables 8–13 and Figs 3–6 show the ratio of EMG amplitude between two specified muscles paired for analysis of the recruitment pattern based on their functional roles.

**Lumbar flexor to lumbar extensor ratio.** *Ipsilateral IO:Ipsilateral MF* – No significant overall time effect, between-group difference and time-and-group interaction effect was detected in both left and right stance phase. Only the BW

**Table 6. Pain intensity (NPRS) in forward flexion standard movement, and performance in Lumbar Movement Control Tests at pre- and post-walking training.**

| Time | BW Group | | FW Group | | Time Effect | Between-group Difference | Time-and-group Interaction |
|---|---|---|---|---|---|---|---|
| | Mean (SD) | Comparison with pre | Mean (SD) | Comparison with pre | p-value (Observed Power) | p-value (Observed Power) | p-value (Observed Power) |
| **Pain intensity (by NPRS)** | | | | | | | |
| Pre | 3.69 (1.20) | – | 2.86 (1.70) | – | <.001* (.98) | .06 (.46) | .45 (.11) |
| Post | 2.75 (1.24) | .014* | 1.79 (1.19) | .026 | | | |
| **LMC test score** | | | | | | | |
| Pre | 3.44 (0.51) | – | 3.43 (.51) | – | <.001* (.96) | .13 (.33) | .19 (.26) |
| Post | 2.94 (0.77) | .006* | 3.14 (.77) | .04 | | | |

*: Bonferroni correction: p<0.025.

**Table 7. Comparison of pre-and-post-15-minute walking training of the performance in respective Lumbar Movement Control Test.**

| | BW Group (n = 16) | | FW Group (n = 14) | | p-value |
|---|---|---|---|---|---|
| | Improved | Not improved | | Improved | Not improved |
| **Standing lumbar flexion** | 2 | 14 | 2 | 12 | .901 |
| **Chest drop** | 0 | 16 | 1 | 13 | .447 |
| **Sitting knee extension** | 3 | 13 | 0 | 14 | <.001 |
| **Rocking backward** | 3 | 13 | 1 | 13 | .185 |

group showed a significant increase in left IO:left MF during the twelfth minute backward walking training (p<0.001) (Table 8).

For the left swing phase, significant between-group differences (p=0.026) and time-and-group interaction effect (p=0.004) were yielded after adjustment for gender. Contrast analysis showed significant time-and-group interaction effect (p=0.004) during the twelfth minute of walking training. For the right swing phase, a significant between-group difference was detected (p=0.022) but no significant differences were found in post-hoc analysis. There was also significant time-and-group interaction observed (p=0.022) with contrast analysis being insignificant. The BW group demonstrated an insignificant decreasing trend in right IO:right MF ratio with a significant reduction detected during the third minute of walking training (p=0.005), on the contrary, the FW group showed an insignificant increasing trend.

*Ipsilateral ES:Ipsilateral RA* – For left ES:left RA during left stance phase, a significant between-group difference (p=0.049) was detected with no significant results obtained in post-hoc tests (Table 9). Also, a nearly significant time effect was observed (p=0.062). There was no statistically significant result yielded in right stance phase. A significant time effect was yielded in left swing phase (p=0.022). A significant reduction in the ratio of FW group in left swing phase during the twelfth minute of walking training (p=0.002) with a significant between-group difference detected (p=0.031). The post-hoc analysis revealed a nearly significant result during the twelfth minute of walking training (p=0.009). For the right swing phase, significant between-group difference (p=0.024) with insignificant post-hoc analysis was found. There was a significant time-and-group effect detected (p=0.009) with contrast analysis showing occurrence of significant interaction during the third (p=0.002) and sixth (p=0.008) minute of walking training.

**Deep stabilizing muscles to superficial muscles ratio.** *Ipsilateral IO:Ipsilateral RA* – No significant time effect, between-group difference or time-and-group interaction was found in left stance and swing phase, and right stance phase (Table 10).

**Table 8. Ratio of electromyographic amplitude between ipsilateral IO and ipsilateral MF (deep muscle pair).**

| Time Point | | BW Group | FW Group | Time Effect | Between-group Difference | | Time-and-Group Interaction | |
|---|---|---|---|---|---|---|---|---|
| | | Mean (SD) | Mean (SD) | p-value (Observed Power) | p-value (Observed Power) | | p-value (Observed Power) | |
| **Left internal oblique: Left multifidus** | | | | | | | | |
| **Left Stance Phase** | Pre | 235.54 (146.12) | 201.78 (121.49) | 0.749 (0.193) | 0.977 (0.050) | – | 0.111 (0.612) | |
| | Post | 244.11 (249.56) | 225.57 (227.19) | | | | | |
| | 3rd min | 315.08 (218.47) | 385.69 (649.19) | | | | | |
| | 6th min | 372.77 (278.73) | 309.51 (367.49) | | | | | |
| | 9th min | 325.23 (178.88) | 275.57 (282.21) | | | | | |
| | 12th min | 371.85 (197.18) [a] | 245.86 (238.00) | | | | | |
| **Left Swing Phase ^** | Pre | 303.68 (275.25) | 206.87 (179.84) | 0.144 (0.568) | 0.026 (0.623) | 0.271 | 0.004 (0.919) | – |
| | Post | 237.03 (289.12) | 339.61 (322.47) | | | 0.259 | | 0.072 |
| | 3rd min | 193.67 (211.15) | 399.11 (512.15) | | | 0.29 | | 0.01 |
| | 6th min | 253.77 (226.79) | 515.59 (578.19) | | | 0.309 | | 0.01 |
| | 9th min | 259.78 (275.68) | 458.89 (515.70) | | | 0.152 | | 0.022 |
| | 12th min | 225.88 (218.57) | 502.48 (470.27) | | | 0.18 | | 0.004 |
| **Right internal oblique: Right multifidus** | | | | | | | | |
| **Right Stance Phase** | Pre | 207.56 (152.64) | 211.08 (209.10) | 0.934 (0.112) | 0.966 (0.050) | – | 0.991 (0.073) | |
| | Post | 239.53 (198.33) | 226.45 (186.40) | | | | | |
| | 3rd min | 227.49 (107.41) | 239.24 (252.52) | | | | | |
| | 6th min | 301.68 (326.19) | 253.31 (214.78) | | | | | |
| | 9th min | 264.92 (173.73) | 252.57 (282.18) | | | | | |
| | 12th min | 243.93 (210.51) | 216.19 (189.59) | | | | | |
| **Right Swing Phase** | Pre | 465.34 (384.55) | 184.12 (145.54) | 0.498 (0.307) | 0.022 (0.808) | 0.016 | 0.022 (0.808) | – |
| | Post | 391.04 (390.84) | 324.83 (360.08) | | | 0.64 | | 0.108 |
| | 3rd min | 136.24 (89.49) [b] | 517.99 (774.82) | | | 0.06 | | 0.024 |
| | 6th min | 180.94 (154.84) | 262.34 (208.06) | | | 0.23 | | 0.018 |
| | 9th min | 217.61 (167.35) | 334.78 (407.95) | | | 0.3 | | 0.055 |
| | 12th min | 212.86 (241.06) | 263.02 (247.16) | | | 0.58 | | 0.026 |

^: Result based on ANCOVA with gender as a covariate

a: Significant changes compared with pre-training assessment, p = 0.000 (Bonferroni correction: p < 0.0083)

b: Significant changes compared with pre-training assessment, p = 0.005 (Bonferroni correction: p < 0.0083)

*Ipsilateral IO:Ipsilateral EO* – Significant time effect (p = 0.046) and between-group difference (p = 0.027) were found in left stance phase. However, contrast and post-hoc analysis showed no significant results. A significant time effect was found in the right stance phase (p = 0.024). Although post-hoc analysis showed insignificant results, both BW and FW groups demonstrated an increasing trend in right IO:right EO ratio, with greater increment observed in the FW group. For the left swing phase, ANCOVA with gender adjustment was adopted for data analysis in which a significant between-group difference was detected (p = 0.036) with insignificant post-hoc analysis. For the right swing phase, a nearly significant time effect was yielded (p = 0.069). An insignificant decreasing trend in right IO:right EO ratio was observed in the BW group, while the direction of trend reversed in the FW group (Table 11).

*Ipsilateral MF:Ipsilateral ES* ratio – Both maximum tolerable gait speed of participants and gender have confounding effects on left MF:ES ratio during left stance phase. The time effect was insignificant with maximum tolerable gait speed being adjusted for analysis. A significant reduction was demonstrated in left MF:left ES ratio of the FW group during the sixth-minute training

**Table 9. Ratio of electromyographic amplitude between ipsilateral ES and ipsilateral RA (superficial muscle pair).**

| Time Point | | BW Group | FW Group | Time Effect | | Between-group Difference | | Time-and-Group Interaction | |
|---|---|---|---|---|---|---|---|---|---|
| | | Mean (SD) | Mean (SD) | p-value (Observed Power) | | p-value (Observed Power) | | p-value (Observed Power) | |
| **Left erector spinae: Left rectus abdominus** | | | | | | | | | |
| **Left Stance Phase** | Pre | 412.18 (254.81) | 279.32 (174.36) | 0.062 (0.696) | | 0.049 (0.512) | 0.11 | 0.752 (0.192) | |
| | Post | 449.27 (355.29) | 441.99 (489.85) | | | | 0.96 | | |
| | 3rd min | 371.78 (253.89) | 243.22 (172.39) | | | | 0.12 | | |
| | 6th min | 397.30 (309.79) | 321.59 (263.18) | | | | 0.48 | | |
| | 9th min | 299.35 (319.95) | 282.90 (203.34) | | | | 0.87 | | |
| | 12th min | 375.34 (354.52) | 348.75 (220.34) | | | | 0.81 | | |
| **Left Swing Phase** | Pre | 601.74 (321.31) | 523.72 (434.25) | 0.022 (0.768) | – | 0.031 (0.691) | 0.577 | 0.071 (0.616) | |
| | Post | 425.58 (408.56) | 464.03 (584.11) | | 0.253 | | 0.834 | | |
| | 3rd min | 680.99 (586.43) | 527.79 (584.57) | | 0.218 | | 0.481 | | |
| | 6th min | 538.01 (392.10) | 283.01 (252.23) | | 0.082 | | 0.047 | | |
| | 9th min | 407.97 (333.33) | 373.00 (366.44) | | 0.018 | | 0.786 | | |
| | 12th min | 684.46 (497.30) | 266.41 (275.54) [a] | | 0.282 | | 0.009 | | |
| **Right erector spinae: Right rectus abdominus** | | | | | | | | | |
| **Right Stance Phase** | Pre | 856.56 (2147.78) | 202.93 (168.62) | 0.674 (0.225) | | 0.309 (0.170) | – | 0.969 (0.092) | |
| | Post | 825.41 (1768.19) | 265.44 (385.93) | | | | | | |
| | 3rd min | 952.79 (2335.19) | 298.75 (229.54) | | | | | | |
| | 6th min | 850.56 (1915.77) | 235.63 (246.74) | | | | | | |
| | 9th min | 797.99 (1818.79) | 254.61 (213.47) | | | | | | |
| | 12th min | 941.66 (2489.68) | 302.41 (292.69) | | | | | | |
| **Right Swing Phase** | Pre | 394.27 (391.37) | 450.28 (330.58) | 0.199 (0.509) | | 0.024 (0.636) | 0.678 | 0.009 (0.877) | – |
| | Post | 438.00 (407.02) | 386.37 (401.80) | | | | 0.729 | | 0.171 |
| | 3rd min | 729.55 (424.76) [b] | 353.46 (415.11) | | | | 0.021 | | 0.002 |
| | 6th min | 720.58 (539.54) | 384.75 (398.53) | | | | 0.066 | | 0.008 |
| | 9th min | 515.17 (442.37) | 380.79 (317.75) | | | | 0.354 | | 0.145 |
| | 12th min | 618.58 (467.09) | 301.57 (407.30) | | | | 0.059 | | 0.009 |

[a]: Significant changes compared with pre-training assessment, p=0.002 (Bonferroni correction: p<0.0083).

[b]: Significant changes compared with pre-training assessment, p=0.004 (Bonferroni correction: p<0.0083).

interval (p=0.007). With adjustment for gender, significant time-and-group interaction effect was detected (p=0.037). Contrast analysis indicated that a significant interaction occurred during the sixth (p=0.007) and twelfth (p=0.006) minute of walking training. No significant results were observed in the right stance phase, left swing phase and right swing phase (Table 12).

**Lumbar extensor to hip extensor ratio.** *Ipsilateral ES:Ipsilateral GMax* - No significant findings were detected during left and right stance phase (Table 13). For the left swing phase, a significant time effect was detected (p<0.001). Contrast analysis detected significant changes comparing initial and final walking assessment (p=0.006) with the BW group showing reduction of greater extent. Significant between-group difference (p=0.006) and time-and-group (p=0.017) interaction were also found in the left stance phase with insignificant post-hoc and contrast analysis. Both gender and maximum tolerable gait speed of participants demonstrated confounding effect to the right ES:right GMax ratio in right swing phase. No significant time effect or time-and-group interaction effect was shown. The right ES:right GMax ratio of the BW group significantly increased during the third minute training interval (p=0.002), while that of the FW group significantly decreased during the twelfth minute training interval (p=0.007). A significant between-group difference was

**Table 10. Ratio of electromyographic amplitude between ipsilateral IO and ipsilateral RA (deep and superficial anterior truncal muscle pair).**

| Time Point | | BW Group | FW Group | Time Effect | Between-group Difference | Time-and-Group Interaction |
|---|---|---|---|---|---|---|
| | | Mean (SD) | Mean (SD) | p-value (Observed Power) | p-value (Observed Power) | p-value (Observed Power) |
| **Left internal oblique: Left rectus abdominus** | | | | | | |
| **Left Stance Phase** | Pre | 953.12 (667.61) | 929.24 (892.21) | 0.553 (0.290) | 0.165 (0.281) | 0.130 (0.586) |
| | Post | 1126.75 (951.58) | 1079.08 (954.55) | | | |
| | 3rd min | 1279.37 (1098.90) | 774.78 (755.16) | | | |
| | 6th min | 1211.53 (977.77) | 879.10 (888.30) | | | |
| | 9th min | 1041.93 (927.84) | 846.30 (744.70) | | | |
| | 12th min | 1267.06 (998.96) | 862.56 (773.44) | | | |
| **Left Swing Phase** | Pre | 1261.16 (1264.32) | 790.30 (801.30) | 0.319 (0.412) | 0.297 (0.177) | 0.203 (0.505) |
| | Post | 1012.75 (815.75) | 735.64 (657.68) | | | |
| | 3rd min | 1228.48 (1248.92) | 1033.28 (957.74) | | | |
| | 6th min | 1179.58 (1116.34) | 909.83 (1084.08) | | | |
| | 9th min | 988.96 (1003.74) | 1105.81 (1288.62) | | | |
| | 12th min | 1161.40 (1137.50) | 873.54 (915.81) | | | |
| **Right internal oblique: Right rectus abdominus** | | | | | | |
| **Right Stance Phase** | Pre | 685.96 (372.69) | 579.75 (578.53) | 0.367 (0.380) | 0.219 (0.229) | 0.722 (0.205) |
| | Post | 967.53 (1077.06) | 627.61 (705.93) | | | |
| | 3rd min | 931.25 (625.43) | 636.90 (625.82) | | | |
| | 6th min | 827.27 (577.16) | 637.11 (703.01) | | | |
| | 9th min | 840.20 (765.88) | 611.88 (713.23) | | | |
| | 12th min | 727.70 (517.96) | 581.72 (577.22) | | | |
| **Right Swing Phase** | Pre | 616.89 (327.35) | 619.39 (783.02) | 0.878 (0.139) | 0.232 (0.218) | 0.183 (0.524) |
| | Post | 690.66 (534.66) | 551.47 (673.30) | | | |
| | 3rd min | 651.19 (405.76) | 399.30 (276.43) | | | |
| | 6th min | 806.63 (782.25) | 404.96 (312.29) | | | |
| | 9th min | 597.21 (468.07) | 511.63 (396.68) | | | |
| | 12th min | 627.52 (495.47) | 724.81 (1078.41) | | | |

demonstrated during the third minute of training interval (p = 0.002), while an insignificant overall between-group difference was found after adjustment for gender.

## Discussion

### Real-time enhancement in activation of deep stabilizing muscles

The activation of lumbar MF suggested that backward walking exercise may not only facilitate lumbar lordosis, but more importantly with segmental control. The EMG amplitude ratio of MF to ES exhibited an increasing trend in the BW group, showing a relatively greater contribution from MF than ES. Meanwhile, the FW group consistently demonstrated significant real-time reduction in MF to ES ratio in the stance phase. These findings suggested that MF as a strong dynamic stabilizer of the lumbar spine contributes greater to the increase in lumbar lordosis after backward walking [48], when compared to superficial muscle (ES). This finding concurred with the previous report of the positive correlation between lumbar lordotic angle and lumbar multifidus activation [49]. This result may be attributed to the toe-heel pattern gait, which requires greater activity of deep back stabilizer and induces more extension movement in lumbar spine [20]. For the profound

Table 11. Ratio of electromyographic amplitude between ipsilateral internal oblique and ipsilateral external oblique (anterior core muscle pair).

| Time Point | | BW Group | FW Group | Time Effect | | Between-group Difference | | Time-and-Group Interaction |
|---|---|---|---|---|---|---|---|---|
| | | Mean (SD) | Mean (SD) | p-value (Observed Power) | | p-value (Observed Power) | | p-value (Observed Power) |
| **Left internal oblique: Left external oblique** | | | | | | | | |
| **Left Stance Phase** | Pre | 271.82 (181.87) | 204.36 (97.02) | 0.046 (0.733) | – | 0.027 (0.617) | 0.21 | 0.598 (0.259) |
| | Post | 296.00 (236.56) | 235.80 (152.28) | | 0.271 | | 0.422 | |
| | 3rd min | 295.49 (164.17) | 197.62 (112.23) | | 0.586 | | 0.065 | |
| | 6th min | 342.67 (235.99) | 225.22 (113.85) | | 0.028 | | 0.091 | |
| | 9th min | 290.42 (171.47) | 219.07 (113.04) | | 0.741 | | 0.185 | |
| | 12th min | 340.16 (223.76) | 237.91 (169.44) | | 0.095 | | 0.174 | |
| **Left Swing Phase ^** | Pre | 308.98 (260.22) | 172.06 (84.28) | 0.854 (0.150) | | 0.036 (0.567) | 0.07 | 0.453 (0.330) |
| | Post | 280.90 (179.15) | 171.54 (114.91) | | | | 0.6 | |
| | 3rd min | 283.18 (234.22) | 255.80 (279.54) | | | | 0.77 | |
| | 6th min | 300.15 (213.47) | 243.93 (181.68) | | | | 0.45 | |
| | 9th min | 250.81 (222.74) | 265.58 (208.11) | | | | 0.85 | |
| | 12th min | 316.16 (319.85) | 175.11 (95.06) | | | | 0.12 | |
| **Right internal oblique: Right external oblique** | | | | | | | | |
| **Right Stance Phase** | Pre | 218.65 (215.53) | 273.01 (446.31) | 0.024 (0.803) | – | 0.981 (0.083) | – | 0.897 (0.052) |
| | Post | 233.67 (267.84) | 339.51 (613.50) | | 0.114 | | | |
| | 3rd min | 279.43 (229.96) | 371.05 (581.43) | | 0.078 | | | |
| | 6th min | 238.64 (172.29) | 335.14 (586.93) | | 0.066 | | | |
| | 9th min | 269.96 (211.56) | 333.46 (544.69) | | 0.016 | | | |
| | 12th min | 258.69 (184.78) | 290.42 (439.54) | | 0.121 | | | |
| **Right Swing Phase** | Pre | 203.70 (121.65) | 238.05 (472.49) | 0.069 (0.682) | | 0.755 (0.061) | – | 0.692 (0.217) |
| | Post | 193.20 (239.01) | 308.66 (542.91) | | | | | |
| | 3rd min | 193.72 (183.69) | 532.32 (1284.30) | | | | | |
| | 6th min | 206.36 (185.95) | 377.59 (680.85) | | | | | |
| | 9th min | 176.17 (194.93) | 337.42 (580.81) | | | | | |
| | 12th min | 191.93 (150.09) | 383.18 (571.24) | | | | | |

^: Result based on ANCOVA with gender as a covariate

fatigability, reduced activation and altered firing timing in MF of LBP patients reported previously [50–52], backward walking exercise can possibly be used to promote or restore the functional capacity of MF in these patients.

Furthermore, backward walking generally recruited more stabilizers than forward walking. Besides lumbar MF as mentioned above, IO also demonstrated greater activation in the BW group. The BW group also exhibited a tendency of increase in amplitude ratio of IO to RA, while the FW group showed the contrary. This opposite tendency shown between two training groups indicated the unique activation strategy of deep stabilizers associated with backward walking. The greater functional demand required to overcome the postural instability and visual cue deficiency in backward walking when compared to forward walking accounts for the upgraded activation strategies of the deep stabilizing muscles [53].

## Real-time change in muscle recruitment pattern during the swing phase

In this study, most of the significant results of EMG analysis were detected in the ipsilateral swing phase, but not the stance phase of the gait. La Scaleia et al. (2014) compared the muscle activation in different walking directions and found

**Table 12. Ratio of electromyographic amplitude between ipsilateral multifidus and ipsilateral erector spinae (deep and superficial posterior truncal muscle pair).**

| Time Point | | BW Group | FW Group | Time Effect | Between-group Difference | Time-and-Group Interaction | |
|---|---|---|---|---|---|---|---|
| | | Mean (SD) | Mean (SD) | p-value (Observed Power) | p-value (Observed Power) | p-value (Observed Power) | |
| Left multifidus: Left erector spinae | | | | | | | |
| Left Stance Phase | Pre | 124.97 (51.93) | 172.54 (88.18) | 0.730 (0.201) | 0.899 (0.52) | 0.037 (0.758) ^ | – |
| | Post | 176.35 (127.12) | 145.63 (65.86) | | | | 0.025 |
| | 3rd min | 145.15 (80.38) | 132.84 (38.64) | | | | 0.011 |
| | 6th min | 121.22 (67.27) | 115.40 (36.16) a | | | | 0.007 |
| | 9th min | 151.08 (80.67) | 140.95 (83.55) | | | | 0.010 |
| | 12th min | 121.86 (58.49) | 111.66 (40.73) | | | | 0.006 |
| Left Swing Phase | Pre | 90.06 (51.94) | 97.72 (42.76) | 0.635 (0.185) | 0.317 (0.166) | 0.681 (0.168) | |
| | Post | 104.87 (56.89) | 100.58 (80.77) | | | | |
| | 3rd min | 123.16 (59.22) | 112.32 (62.07) | | | | |
| | 6th min | 115.04 (76.80) | 88.22 (36.39) | | | | |
| | 9th min | 134.13 (66.94) | 97.03 (53.33) | | | | |
| | 12th min | 121.46 (121.65) | 78.86 (39.17) | | | | |
| Right multifidus: Right erector spinae | | | | | | | |
| Right Stance Phase | Pre | 152.14 (82.58) | 169.84 (83.22) | 0.173 (0.535) | 0.707 (0.066) | 0.753 (0.192) | |
| | Post | 133.64 (66.48) | 141.03 (96.05) | | | | |
| | 3rd min | 123.05 (63.19) | 118.73 (46.50) | | | | |
| | 6th min | 121.14 (55.09) | 140.09 (71.85) | | | | |
| | 9th min | 123.70 (61.67) | 132.73 (79.60) | | | | |
| | 12th min | 111.82 (49.78) | 111.15 (58.74) b | | | | |
| Right Swing Phase | Pre | 86.29 (45.17) | 78.73 (33.91) | 0.917 (0.121) | 0.416 (0.125) | 0.758 (0.190) | |
| | Post | 93.07 (76.69) | 98.38 (84.19) | | | | |
| | 3rd min | 83.99 (42.28) | 89.18 (72.17) | | | | |
| | 6th min | 98.40 (62.17) | 101.78 (75.08) | | | | |
| | 9th min | 92.88 (48.81) | 90.32 (61.33) | | | | |
| | 12th min | 83.00 (38.96) | 112.96 (45.68) | | | | |

^: Results based on ANCOVA with gender as a covariate.

a: Significant changes compared with pre-training assessment, p = 0.007 (Bonferroni correction: p < 0.0083).

b: Significant changes compared with pre-training assessment, p = 0.008 (Bonferroni correction: p < 0.0083).

some differences in the recruitment patterns of ES in forward and backward walking [41]. The activity of ES during the swing phase in BW is higher than that in FW. As the ipsilateral lower limb performs hip and lumbar extension in BW, more recruitment in extensor muscles is likely to be required. On the contrary, in the stance phase, spikes of ES activity were observed during the pre-swing phase in both forward and backward walking. Therefore, the real-time changes in muscle recruitment patterns, especially ES-related muscle pairings, were detected mostly in the swing phase instead of the stance phase.

## The effect of gait speed on muscle recruitment patterns

Significant differences in gait speed at baseline and during training were observed between the BW and FW groups. The study adopted a few methods to minimize the effect of gait speed on muscle recruitment patterns. First, to account for

**Table 13. Ratio of electromyographic amplitude between ipsilateral erector spinae and ipsilateral gluteus maximus (ipsilateral lumbopelvic muscle pair).**

| Time Point | | BW Group | FW Group | Time Effect | | Between-group Difference | | Time-and-Group Interaction | |
|---|---|---|---|---|---|---|---|---|---|
| | | Mean (SD) | Mean (SD) | p-value (Observed Power) | | p-value (Observed Power) | | p-value (Observed Power) | |
| **Left erector spinae: Left gluteus maximus** | | | | | | | | | |
| **Left Stance Phase** | Pre | 106.87 (95.13) | 70.80 (46.65) | 0.919 (0.089) | | 0.531 (0.094) | – | 0.597 (0.197) | |
| | Post | 76.72 (68.06) | 70.71 (35.07) | | | | | | |
| | 3rd min | 109.39 (120.81) | 72.44 (48.59) | | | | | | |
| | 6th min | 94.93 (72.50) | 84.94 (72.74) | | | | | | |
| | 9th min | 91.96 (98.65) | 81.33 (58.12) | | | | | | |
| | 12th min | 91.80 (92.52) | 94.20 (62.98) | | | | | | |
| **Left Swing Phase** | Pre | 658.46 (567.81) | 419.76 (351.50) | 0.000 (0.986) | – | 0.006 (0.813) | 0.19 | 0.017 (0.830) | – |
| | Post | 351.00 (364.21) | 264.67 (259.83) | | 0.006 | | 0.47 | | 0.239 |
| | 3rd min | 797.82 (883.03) | 336.12 (333.18) | | 0.265 | | 0.08 | | 0.119 |
| | 6th min | 469.16 (408.57) | 270.14 (319.48) | | 0.074 | | 0.15 | | 0.867 |
| | 9th min | 484.77 (408.77) | 262.55 (390.16) | | 0.068 | | 0.14 | | 0.763 |
| | 12th min | 555.50 (462.68) | 233.61 (212.20) | | 0.307 | | 0.02 | | 0.488 |
| **Right erector spinae: Right gluteus maximus** | | | | | | | | | |
| **Right Stance Phase** | Pre | 64.13 (35.55) | 66.74 (63.51) | 0.621 (0.249) | | 0.609 (0.079) | – | 0.673 (0.226) | |
| | Post | 73.37 (50.09) | 70.60 (54.29) | | | | | | |
| | 3rd min | 99.62 (75.65) | 79.59 (54.27) | | | | | | |
| | 6th min | 84.57 (59.39) | 62.98 (45.40) | | | | | | |
| | 9th min | 111.53 (117.25) | 72.76 (55.64) | | | | | | |
| | 12th min | 81.16 (77.39) | 76.89 (37.65) | | | | | | |
| **Right Swing Phase** | Pre | 302.42 (394.06) | 331.43 (204.69) | 0.188 (0.395) | | 0.087 (0.403) ^ | 0.806 | 0.067 (0.579) | |
| | Post | 345.60 (340.22) | 230.66 (228.08) | | | | 0.294 | | |
| | 3rd min | 719.27 (479.91) a | 239.56 (230.42) | | | | 0.002 | | |
| | 6th min | 737.15 (696.46) | 316.06 (290.54) | | | | 0.04 | | |
| | 9th min | 458.97 (476.61) | 315.22 (276.72) | | | | 0.33 | | |
| | 12th min | 478.16 (420.95) | 210.89 (207.07) b | | | | 0.04 | | |

^: Results based on ANCOVA with gender as a covariate.

a: Significant changes compared with pre-training assessment, p=0.002 (Bonferroni correction: p<0.0083).

b: Significant changes compared with pre-training assessment, p=0.007 (Bonferroni correction: p<0.0083).

the significantly higher baseline maximum tolerable gait speed in the BW group, ANCOVA was performed with gait speed as a covariate. Additionally, the mean gait speed during BW training was significantly lower than FW, which is possibly contributed by the increased difficulty of BW and the lack of visual cues for progression [53]. Previous research suggests that gait speeds differing from self-selected speeds may affect both activity levels and muscle activation patterns [54]. Therefore, participants were instructed to walk at a comfortable, self-selected pace without external assistance to ensure comparable exertion levels between groups.

Moreover, studies supported that with increased walking speed, mean EMG amplitudes increased generally while phase dependent muscle activation patterns remained similar [55]. Neptune et al. (2007) demonstrated that while EMG amplitudes increase with walking speed (0.4–2.0 m/s), the phasing and functional roles of gluteus maximus and lower limb muscle activations remain invariant, suggesting that relative coordination patterns are robust to speed changes [56].

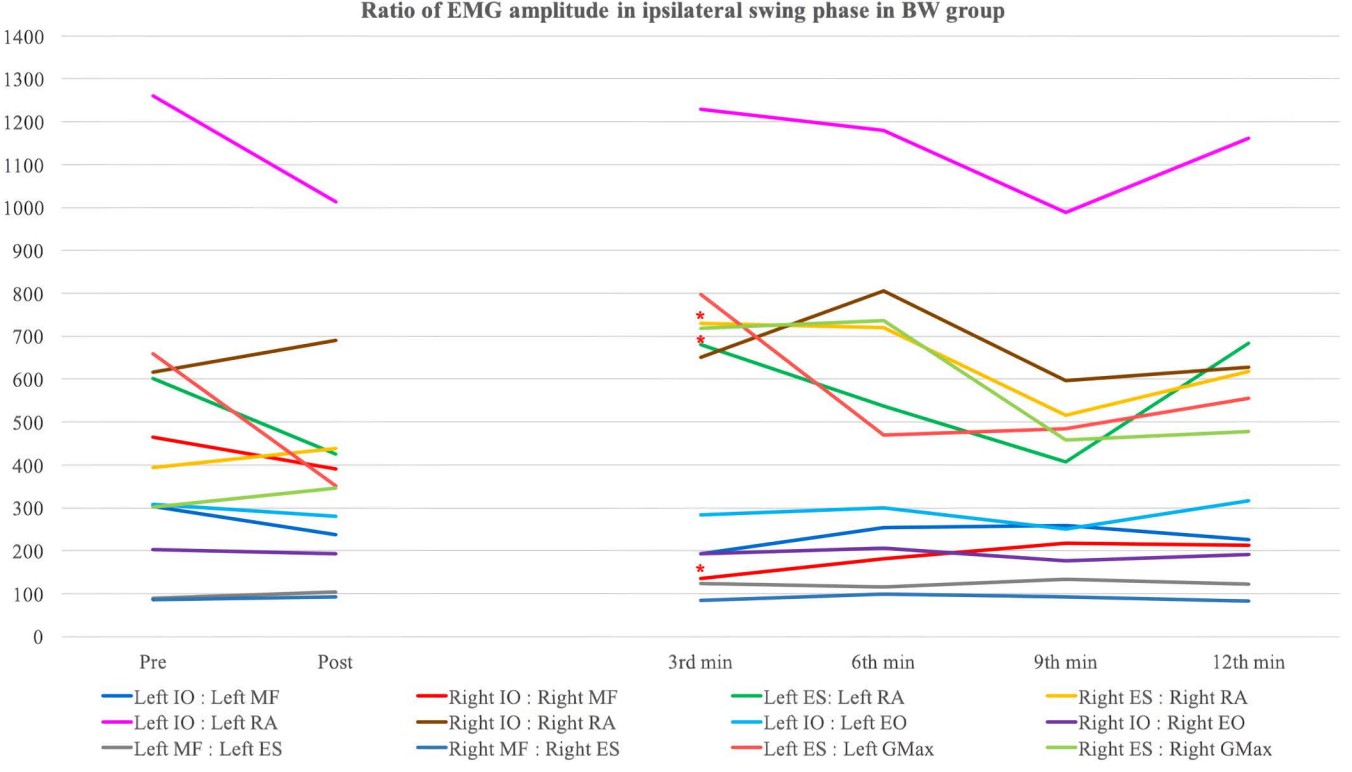

**Fig 3. The ratio of EMG amplitude in ipsilateral swing phase in the backward walking group.** *: Significant changes compared with pre-training assessment (Bonferroni correction: p < 0.0083).

Cappellini et al. (2006) investigated back extensors, abdominal and gluteal muscles, suggested stable muscle activation timing during walking across speeds of 0.83–3.33 m/s, with differences primarily in amplitude rather than pattern [57]. By analyzing EMG in ratios based on their functional roles, the impact of speed-induced amplitude variations was minimized. Thus, revealing muscle recruitment patterns, ensuring that the observed differences in muscle recruitment are primarily attributable to walking direction, and enhancing the validity of our findings.

## Immediate post-training improvement in LMC test performance

Present results suggest that backward walking helps improve motor control in lumbar flexion direction. The number of incorrect LMC tests reduced significantly in participants who completed backward walking training. The improved performance of the BW group displayed in sitting knee extension and rocking backward nearly reached a significant level. The improvement in LMC might be attributed to the enhancement in muscle activity of lumbar extensors relative to gluteal muscles. The significant increase in amplitude ratio of right ES:right GMax of the BW group during the right swing phase suggests an augmented relative activity of lumbar extensor. Higher lumbar extensor activity enhances the relative stiffness of the trunk against flexion torque, contributing to better motor control of the flexion movement test [5]. Among the four LMC tests, only the sitting knee extension test showed a significant time-and-group interaction effect. Unlike the other tests, sitting knee extension assesses the ability of participants to maintain a neutral position of the lumbar spine while performing an open-kinetic chain lower limb movement, which is similar to the demand associated with the suspended lower limb movement during the swing phase. The similarity in movement demands acquired between sitting knee extension and swing phase explains the unique time-and-group intervention effect demonstrated in this LMC test.

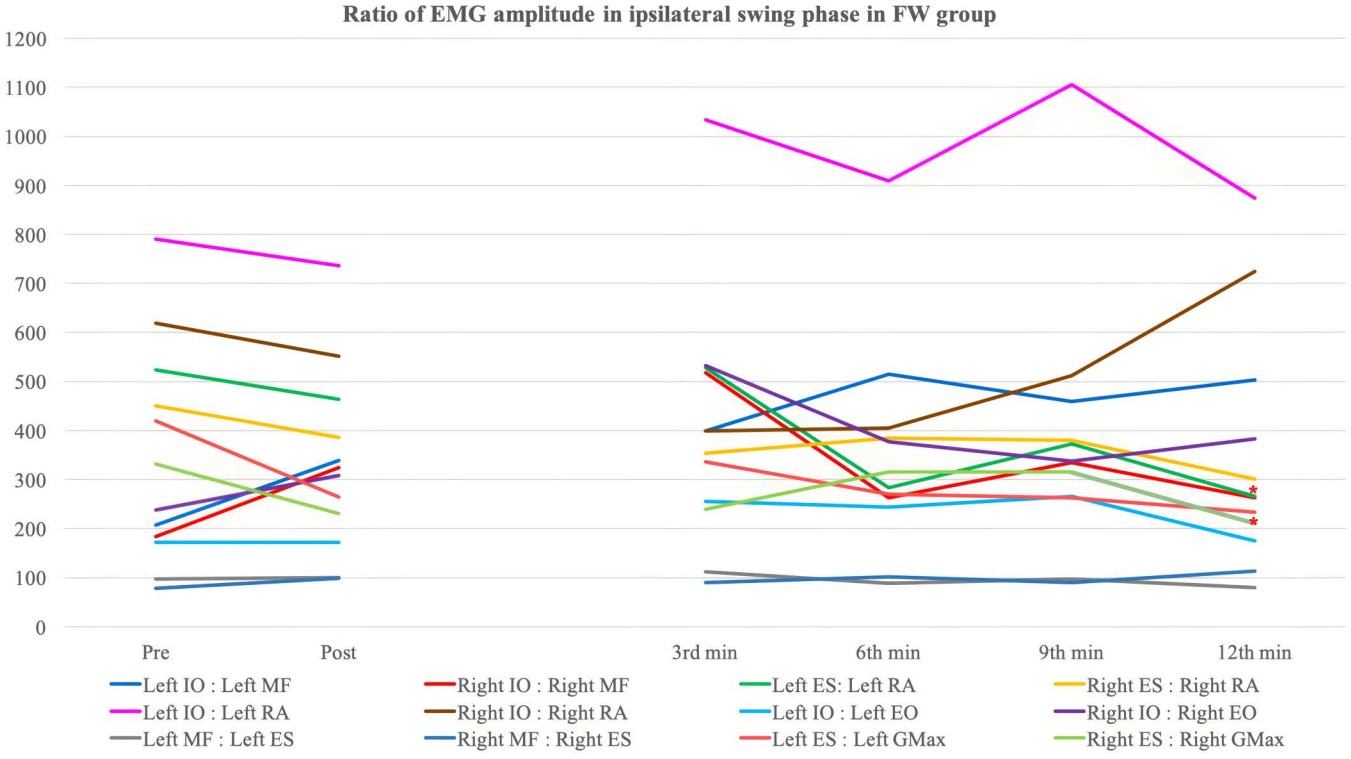

**Fig 4. The ratio of EMG amplitude in ipsilateral swing phase in the forward walking group.** *: Significant changes compared with pre-training assessment (Bonferroni correction: p < 0.0083).

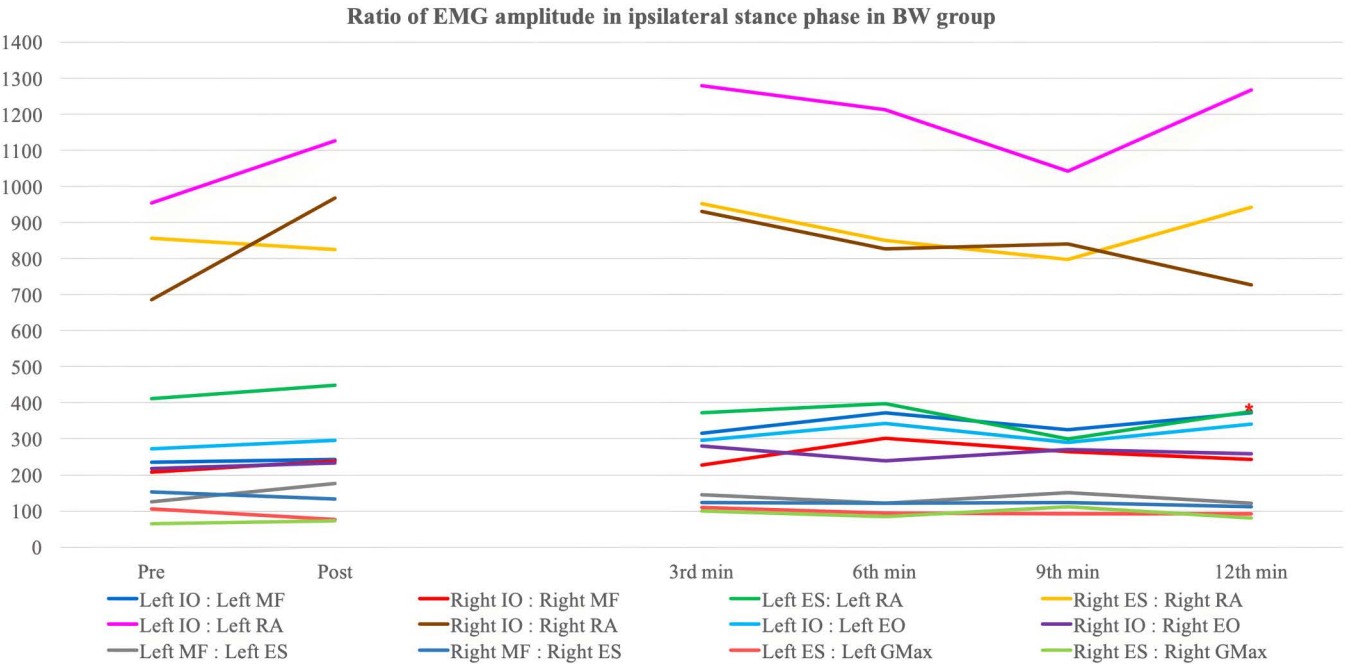

**Fig 5. The ratio of EMG amplitude in ipsilateral stance phase in the backward walking group.** *: Significant changes compared with pre-training assessment (Bonferroni correction: p < 0.0083).

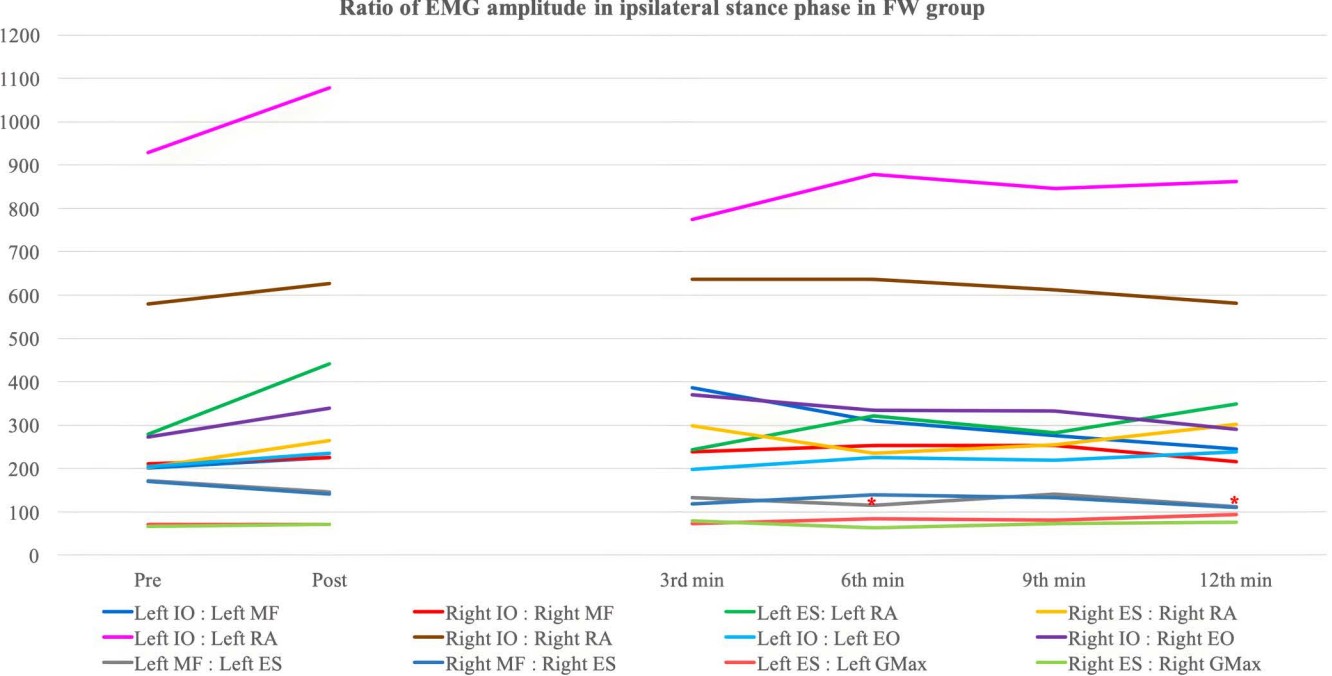

**Fig 6. The ratio of EMG amplitude in ipsilateral stance phase in the forward walking group.** *: Significant changes compared with pre-training assessment (Bonferroni correction: p < 0.0083).

Unlike isolated lumbar motor control training, backward walking training in this study attempted to enhance the activation of deep stabilizing muscles in a functional position. According to the motor learning model proposed by O'Sullivan, patients have to train the local muscle system functionally in order to reach the autonomous stage, which enables patients to activate their deep stabilizing muscles with a low degree of attention during daily functional tasks [58]. Since training in functional position is of utmost importance in lumbar motor control training, backward walking training with duration as short as 15 minutes could elicit a significant improvement in flexion lumbar motor control immediately after training.

### Relatively low reliability of the right erector spinae EMG measurement

Although the level of reliability of right erector spinae EMG measurement is considered to be moderate with an ICC value of 0.558, it is indeed the lowest amongst the listed muscles. Therefore, interpretation of results involving EMG analysis of the right erector spinae would require extra caution irrespective to significant and insignificant findings statistically, i.e., the significant increase in right ES:right GMax activity ratio during the 3rd minute of BW training and significant decrease in the same ratio found during the 12th minute of FW training, and those statistically insignificant comparisons reported in Tables 9, 12 and 13. There are few possible reasons contributing to the relatively lower reliability in right erector spinae measurement. Hao et al. proposed that chronic low back pain can result in asymmetric activity of the erector spinae, with some individuals showing more variable or less reliable EMG signals on one side [59]. Lumbar movement control (LMC) tests selected in this study detect the changes in movement control of the lumbar spine in the sagittal plane, but not that in the transverse or frontal plane. Besides, the movements involved in the selected tests are symmetrical, resulting in a lack of ability to differentiate lumbar flexion-rotation syndrome from lumbar flexion syndrome [28]. Patients with lumbar flexion-rotation syndrome are presented with unilateral change in rotation LMC [5]. The possible inclusion of patients with lumbar flexion-rotation syndrome may further exaggerate the increase in variability of erector spinae activity on one side.

Moreover, research found that rectus abdominis and external oblique, in comparison to erector spinae, demonstrated relatively constant patterns of activity throughout the stride cycle of gait [60]. This suggested that the erector spinae may present with a greater variability in muscle activity during walking. Lastly, the magnified crosstalk effect may contribute to the relatively low ICC as erector spinae lay deep to several muscle layers [61]. It is respected that the moderate reliability of right erector spinae measurement may be attributed to the variation of erector spinae activity in dynamic movement based on the above reasons.

## Clinical implications

This study provides some emerging evidence to substantiate the use of backward walking in improving performance of LMC tests for individuals with lumbar flexion syndrome. From the demographics and baseline assessment, the results of this study could only be generalized to those NSLBP individuals with clinical features of flexion syndrome without no radiating symptoms, low pain intensity, and low disability levels. With respect to the heterogeneity of the entire spectrum of LBP population, cautions are required when interpreting the generalizability and application of the present findings in subgroups other than that specified in this study. Furthermore, the effect of backward walking on patients of moderate and high severity requires investigation by future studies.

This study found significant real-time changes in activity of lumbar extensor relative to hip extensors during a single session of 15-minute backward walking training without any adverse event, providing an emerging evidence to substantiate the potential benefits of backward walking as a treatment option for patients with lumbar flexor syndrome. However, most of these kinematic changes were no longer detected in the final walking assessment immediately after backward walking training. To induce long-lasting effects, exercise and motor control training are commonly conducted for four to six weeks. Previous studies suggested that a four-week backward walking program with a duration of 10–15 minutes and frequency of three to four times per week induced a significant improvement in the lumbar range of motion in sagittal plane [26]. Another randomized controlled trial revealed that a six-week training program focusing on motor control and maintaining a neutral spine during physical activity significantly reduced pain intensity in 6-month and 1-year follow-up periods [62]. A longer training period may be required to transfer the enhanced muscle recruitment pattern or spinal alignment to functional forward walking tasks.

## Limitations

First of all, although statistically significant changes in the aforementioned recruitment patterns (expressed in EMG findings) were reviewed during and after backward walking training, it remains difficult to affirm their potential therapeutic effect on managing LBP due to the marginal effect size or inconsistent activation pattern observed across phases of the gait cycle. Secondly, the coexistence of lumbar flexion syndrome and extension syndrome, or other conditions within the present cohort of participants may influence the results. Previous study showed that 80% of the subjects with NSLBP presented with lumbar movement control impairment in more than one direction [28]. Since backward walking exercise aims at improving the relative stiffness of lumbar extensor, which is a treatment goal specific for improving performance of LMC tests, the therapeutic effect may not be sufficient to induce a clinically meaningful improvement in patients with multi-directional LMC impairment. Future studies are recommended to implement LMC tests targeting other directions as a screening tool, which helps reviewing the genuine effect of backward walking exercise on patients with only lumbar flexion syndrome. Furthermore, the immediate or direct application of findings obtained from the present study will be limited for its relatively small cohort of participants recruited.

## Conclusions

Individuals with lumbar flexion syndrome with low pain and disability level, and no radiating symptoms may potentially benefit from backward walking exercise. A real-time increase in average dynamic lordotic angle with promotion of back extensor and deep stabilizing muscle recruitment pattern, immediate post-training improvement in pain intensity and LMC

test performance were induced by a 15-minute backward walking training. Bonferroni corrections were applied to control for Type I error inflation due to multiple comparisons. As the effect sizes of some EMG changes are marginal, the findings provide insight on the potential benefit of backward walking exercise, while the results should be interpreted with caution. future study with larger sample size and longer training period which help improving the power analyses, and participants with sole lumbar motor control impairment in flexion direction that helps addressing the clinical application are highly recommended to evaluate the true effect of backward walking exercise on lumbar flexion syndrome of the chronic LBP sufferers.

## Supporting information

**S1 Fig. Experimental procedures.**
(TIF)

**S2 Fig. Specified intervals of EMG and spinal alignment measurements during the 15-minute walking training.**
(TIF)

**S3 Fig. The ratio of EMG amplitude in ipsilateral swing phase in the backward walking group.** *: Significant changes compared with pre-training assessment (Bonferroni correction: $p < 0.0083$).
(TIF)

**S4 Fig. The ratio of EMG amplitude in ipsilateral swing phase in the forward walking group.** *: Significant changes compared with pre-training assessment (Bonferroni correction: $p < 0.0083$).
(TIF)

**S5 Fig. The ratio of EMG amplitude in ipsilateral stance phase in the backward walking group.** *: Significant changes compared with pre-training assessment (Bonferroni correction: $p < 0.0083$).
(TIF)

**S6 Fig. The ratio of EMG amplitude in ipsilateral stance phase in the forward walking group.** Significant changes compared with pre-training assessment (Bonferroni correction: $p < 0.0083$).
(TIF)

**S7 File. Supplementary file of datasets.**
(XLSX)

## Author contributions

**Conceptualization:** Ellen Chan, Lok-Yi Chan, Hung-Kit Fong, Yiu-To Mak, Sharon MH Tsang.

**Data curation:** Ellen Chan, Lok-Yi Chan, Hung-Kit Fong, Yiu-To Mak, Sharon MH Tsang.

**Formal analysis:** Ellen Chan, Lok-Yi Chan, Hung-Kit Fong, Yiu-To Mak, Patrick Wai-Hang Kwong, Eliza Rui Sun, Clare CW Yu, Sharon MH Tsang.

**Investigation:** Ellen Chan, Lok-Yi Chan, Hung-Kit Fong, Yiu-To Mak, Patrick Wai-Hang Kwong, Sharon MH Tsang.

**Methodology:** Ellen Chan, Lok-Yi Chan, Hung-Kit Fong, Yiu-To Mak, Patrick Wai-Hang Kwong, Sharon MH Tsang.

**Project administration:** Sharon MH Tsang.

**Resources:** Patrick Wai-Hang Kwong, Eliza Rui Sun, Sharon MH Tsang.

**Software:** Eliza Rui Sun.

**Supervision:** Sharon MH Tsang.

**Validation:** Sharon MH Tsang.

**Writing – original draft:** Ellen Chan, Lok-Yi Chan, Hung-Kit Fong, Yiu-To Mak, Patrick Wai-Hang Kwong, Eliza Rui Sun, Clare CW Yu, Sharon MH Tsang.

**Writing – review & editing:** Ellen Chan, Lok-Yi Chan, Hung-Kit Fong, Yiu-To Mak, Patrick Wai-Hang Kwong, Eliza Rui Sun, Clare CW Yu, Sharon MH Tsang.

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
