## [Decision Letter · Decision Letter 0]

14 Apr 2025

PONE-D-24-59535Real-time and immediate effects of backward walking exercise on pain intensity and lumbopelvic movement control in individuals with chronic non-specific low back pain with lumbar flexion syndromePLOS ONE

Dear Dr.  Tsang,

Thank you for submitting your manuscript to PLOS ONE. After careful consideration, we feel that it has merit but does not fully meet PLOS ONE’s publication criteria as it currently stands. Therefore, we invite you to submit a revised version of the manuscript that addresses the points raised during the review process.

We look forward to receiving your revised manuscript.

Kind regards,

Holakoo Mohsenifar

Academic Editor

PLOS ONE

Journal Requirements:

Reviewers' comments:

Reviewer's Responses to Questions

**Comments to the Author**

1. Is the manuscript technically sound, and do the data support the conclusions?

Reviewer #1: Yes

Reviewer #2: Yes

2. Has the statistical analysis been performed appropriately and rigorously? 

Reviewer #1: Yes

Reviewer #2: Yes

3. Have the authors made all data underlying the findings in their manuscript fully available?

Reviewer #1: Yes

Reviewer #2: No

4. Is the manuscript presented in an intelligible fashion and written in standard English?

Reviewer #1: Yes

Reviewer #2: Yes

5. Review Comments to the Author

Reviewer #1: I congratulate the authors for this detailed and well-researched article.. The low number of cases can be added to the limitations section. No need to see it again, it's suitable for printing. Good luck.

Reviewer #2: This study addresses a novel and under-researched area by investigating the effects of backward walking on neuromuscular control and pain in NSLBP patients with lumbar flexion syndrome. The work is thoughtfully designed and uses appropriate statistical techniques, including adjustment for covariates.

However, several methodological and interpretive issues should be addressed to strengthen the manuscript and improve clarity and transparency. These issues are detailed below.

Major Comments

1) No power analysis is presented to support the adequacy of the sample size. Please clarify whether the study was powered to detect changes in EMG ratios or LMC test outcomes. Given the small size and multiple comparisons, there is a risk of both Type I and Type II errors.

2) Please discuss more on the low generalizability of your results to the entire LBP population, as shwon by your sample characteristics

3) The low reliability of the right erector spinae measurement (ICC = 0.558) raises concern about the robustness of findings involving that muscle group. Please discuss the implications and consider sensitivity analyses excluding low-reliability data. Why didn't you evaluate the reliability for all muscle groups ?

The use of EMG ratios as primary metrics is interesting but not clearly justified. It would help to explain how these ratios translate to meaningful physiological or clinical changes (in terms of directions of the change, for example).

More discussion is needed on the limitations of surface EMG in assessing deep muscle activity (e.g., multifidus) and the potential for crosstalk.

4) Concerning LMC Test Scoring, even if reliability appears to be good, and if the use of video assessment is commendable, but please provide more details on assessor training, blinding, and inter-rater agreement for each test. The binary scoring system for LMC tests (correct/incorrect) may lack sensitivity to subtle improvements. Why didn't you use kinematic measures ?

5) While ANCOVA was used to adjust for this, it’s important to discuss whether gait speed difference (BW vs FW) might still influence muscle recruitment patterns and overall interpretation of group differences.

6) The results are statistically thorough - extensive ANCOVAs, corrections for multiple comparisons, and consideration of covariates like gender and gait speed. However, the high number of comparisons risks Type I error inflation, despite Bonferroni corrections. Moreover, many EMG changes are statistically significant but marginal in effect size or inconsistent across limbs/phases. The interpretation of these changes is at times overly optimistic, and once again I think the conclusions need to be nuanced.

7) The current data availability statement suggests restrictions apply. Please clarify why data aren't available, or if data will be made available upon request or via repository.

8) Authors should carefully also review the abstract in accordance with the necessary

Minor Comments:

1) I think some tenses need to be adjusted throughout the manuscript, that is generally well-written and structured.

2) The extensive EMG data could benefit from clearer visual summaries. Could you include simplified graphs showing key trends ?

3) Please ensure consistent terminology throughout the manuscript (e.g., “flexion LMC performance” vs. “lumbar movement control tests”)

Thank you for the opportunity to review this manuscript.

6. PLOS authors have the option to publish the peer review history of their article (what does this mean? ). If published, this will include your full peer review and any attached files.

**Do you want your identity to be public for this peer review?** For information about this choice, including consent withdrawal, please see our Privacy Policy .

Reviewer #1: **Yes: ** Melda Soysal Tomruk

Reviewer #2: No

---

## [Author Response · Author response to Decision Letter 1]

2 Jul 2025

Manuscript title: Real-time and immediate effects of backward walking exercise on pain intensity and lumbopelvic movement control in individuals with chronic non-specific low back pain with lumbar flexion syndrome

Manuscript ID: PONE-D-24-59535R1

Reviewer #1:

1. I congratulate the authors for this detailed and well-researched article. The low number of cases can be added to the limitations section. No need to see it again, it's suitable for printing. Good luck.

Response: Thank you for your comments and recommendation. We have included the limitation related to the low sample size of our study in the limitation and conclusion of the manuscript (Line 667-670, 687-690), as advised.

Reviewer #2:

1. 1) No power analysis is presented to support the adequacy of the sample size. Please clarify whether the study was powered to detect changes in EMG ratios or LMC test outcomes. Given the small size and multiple comparisons, there is a risk of both Type I and Type II errors.

Response: Thank you for your comment. Power analysis of this study was done for sample size calculation with a medium effect size of f=0.25 for detecting changes in Lumbar Motor Control test as the outcome, using G*Power version 3.1.9.2. We have included the details of power analysis in the methodology section of the manuscript (Line 158-163), as advised.

2. Please discuss more on the low generalizability of your results to the entire LBP population, as shown by your sample characteristics

Response: We have included the discussion on the generalizability of our results in the clinical implication section of the manuscript (Lines 644-651) under “Clinical Implications”, lines 667-670 under “Limitations” of the Discussion section), as advised.

3. The low reliability of the right erector spinae measurement (ICC = 0.558) raises concern about the robustness of findings involving that muscle group. Please discuss the implications and consider sensitivity analyses excluding low-reliability data. Why didn't you evaluate the reliability for all muscle groups ?

Response: The reliability of all six pairs of muscle groups has now been presented in Table 5 and manuscript (Line 387-390) under “Normalized EMG data” of the result section.

We appreciate the reviewer’s concern regarding the moderate reliability of right erector spinae EMG measurement, which acquires relatively lower ICC amongst the listed muscles. Results involving right erector spinae may require interpretation with cautions. There are few possible reasons contributing to the relatively lower reliability in right erector spinae measurement. Hao et al. proposed that chronic low back pain can result in asymmetric activity of the erector spinae, with some individuals showing more variable or less reliable EMG signals on one side [a]. Lumbar movement control (LMC) tests selected in this study detect the changes in movement control of the lumbar spine in the sagittal plane, but not that in the transverse or frontal plane. Besides, the movements involved in the selected tests are symmetrical, resulting in a lack of ability to differentiate lumbar flexion-rotation syndrome from lumbar flexion syndrome [b]. Patients with lumbar flexion-rotation syndrome are presented with unilateral change in rotation LMC [c]. The possible inclusion of patients with lumbar flexion-rotation syndrome may further exaggerate the increase in variability of erector spinae activity on one side. Moreover, research found that rectus abdominis and external oblique, in comparison to erector spinae, demonstrated relatively constant patterns of activity throughout the stride cycle of gait [d]. This suggested that erector spinae may present with greater variability in muscle activity during walking. Lastly, the magnified crosstalk effect may contribute to the relatively low ICC as erector spinae lay deep to several muscle layers [e]. We respect that the moderate reliability of right erector spinae measurement may be attributed to the variation of erector spinae activity in dynamic movement based on the above reasons. Therefore, sensitivity analyses are not adopted to eliminate the subject showing different erector spinae activity.

4. Concerning LMC Test Scoring, even if reliability appears to be good, and if the use of video assessment is commendable, but please provide more details on assessor training, blinding, and inter-rater agreement for each test. The binary scoring system for LMC tests (correct/incorrect) may lack sensitivity to subtle improvements. Why didn't you use kinematic measures ?

Response: The assessor training included the provision of standardized procedures of LMC tests with detailed criteria for movement assessment and scoring system to the assessors. Mock reviews were conducted with trials of LMC tests on healthy subjects and video recording in a standardized environment prior to the data collection. Each assessor’s videos were reviewed, and training was given on reaching inter-rater agreement when differing in evaluations. During data collection, the walking direction of the participant was allocated by drawing a ballot on-site. Therefore, off-site assessors were blinded to the allocated walking direction. They have now been stated in paragraph 2 of the Instrumentation and measurement section (Lines 246-254). Meanwhile, the authors are fully aware of the benefits of using motion capturing for evaluating the control of movements during the tests, however, the use of kinematic measures would require attachment of markers or sensors onto the skin, which might provide tactile feedback on spinal position and thus affect the results of LMC tests. Also, the LMC tests adopted in the study were more readily used for LBP subgroup classification in clinical settings.

5. While ANCOVA was used to adjust for this, it’s important to discuss whether gait speed difference (BW vs FW) might still influence muscle recruitment patterns and overall interpretation of group differences.

Response: We appreciate the reviewer’s concern regarding the potential influence of gait speed differences between BW and FW groups on muscle recruitment patterns. Previous literature stated that mean EMG amplitudes increased with gait speed while phase dependent muscle activation patterns remained similar. We employed ratio-based EMG analysis, focusing on the functional roles of lumbopelvic muscles, so as to reveal the muscle recruitment patterns instead of EMG amplitude. ANCOVA analysis with the walking speed of BW and FW would only be applicable for the comparisons of the real-time changes in EMG amplitude (expressed in percentage of MVC) and co-contraction of paired muscles (expressed in co-contraction index). We have added a discussion of these points in the manuscript’s discussion section (lines 573-592) to clarify the robustness of our approach.

6. The results are statistically thorough - extensive ANCOVAs, corrections for multiple comparisons, and consideration of covariates like gender and gait speed. However, the high number of comparisons risks Type I error inflation, despite Bonferroni corrections. Moreover, many EMG changes are statistically significant but marginal in effect size or inconsistent across limbs/phases. The interpretation of these changes is at times overly optimistic, and once again I think the conclusions need to be nuanced.

Response: Thank you for your comment. To increase the transparency about the potential for type I error inflation, and to advise readers to interpret the results with caution, we have amended the Limitations and Conclusion sections (Lines 667-670, 687-688), as advised.

7. The current data availability statement suggests restrictions apply. Please clarify why data aren't available, or if data will be made available upon request or via repository.

Response: Due to the large volume of dataset from extensive statistical comparison, the current data is available with restrictions. Data could be made available upon request.

8. Authors should carefully also review the abstract in accordance with the necessary revision of the manuscript.

Response: Abstract section has been revised with references to the revisions made to the manuscript detailed above (Abstract section, pages 2-3).

9. Minor comments: Minor Comments:

1) I think some tenses need to be adjusted throughout the manuscript, that is generally well-written and structured.

2) The extensive EMG data could benefit from clearer visual summaries. Could you include simplified graphs showing key trends ?

3) Please ensure consistent terminology throughout the manuscript (e.g., “flexion LMC performance” vs. “lumbar movement control tests”)

Response: We have revised our manuscript by 1) correcting the grammatical errors for some tenses used in our reporting (line 239, 254, 645, 646, 675), 2) including three graphical presentations of our EMG results to enhance the comprehension of our findings on the muscle recruitment pattern during walking (Figure 3-6), and 3) revising the relevant terminology related to the lumbar motor control test and performance throughout the manuscript (line 145, 240, 646).

End of the response letter

---

## [Decision Letter · Decision Letter 1]

18 Jul 2025

PONE-D-24-59535R1Real-time and immediate effects of backward walking exercise on pain intensity and lumbopelvic movement control in individuals with chronic non-specific low back pain with lumbar flexion syndromePLOS ONE

Dear Dr. Tsang,

Thank you for submitting your manuscript to PLOS ONE. After careful consideration, we feel that it has merit but does not fully meet PLOS ONE’s publication criteria as it currently stands. Therefore, we invite you to submit a revised version of the manuscript that addresses the points raised during the review process.

We look forward to receiving your revised manuscript.

Kind regards,

Holakoo Mohsenifar

Academic Editor

PLOS ONE

Journal Requirements:

Reviewers' comments:

Reviewer's Responses to Questions

**Comments to the Author**

1. If the authors have adequately addressed your comments raised in a previous round of review and you feel that this manuscript is now acceptable for publication, you may indicate that here to bypass the “Comments to the Author” section, enter your conflict of interest statement in the “Confidential to Editor” section, and submit your "Accept" recommendation.

Reviewer #2: (No Response)

2. Is the manuscript technically sound, and do the data support the conclusions?

Reviewer #2: Yes

3. Has the statistical analysis been performed appropriately and rigorously? 

Reviewer #2: Yes

4. Have the authors made all data underlying the findings in their manuscript fully available?

Reviewer #2: No

5. Is the manuscript presented in an intelligible fashion and written in standard English?

Reviewer #2: Yes

6. Review Comments to the Author

Reviewer #2: Dear Editors, Dear Authors,

Thank you for the opportunity to re-evaluate the revised manuscript. I appreciate the authors’ thoughtful and detailed responses to my previous comments, as well as the substantive revisions made throughout the manuscript. The additions – including clarification of the sample size calculation, expanded discussion of generalizability, revised figures, and improvements to terminology and grammar – have significantly strengthened the paper.

Most of my concerns have been addressed appropriately. However, I would like to request a few minor revisions prior to recommending acceptance:

- The inclusion of ICCs for all muscle groups is appreciated, and the discussion of the factors influencing the low reliability of the right erector spinae is thorough. However, I suggest that the main text (particularly in the Results and Discussion) include explicit cautionary language wherever findings involving this muscle are interpreted, to ensure that readers are aware of the limitations in reliability.

- While the discussion now notes the potential for Type I error and small effect sizes, I still recommend that the authors slightly temper the language used in key interpretive statements.

- The revised data availability statement notes that data are available upon request due to the dataset’s complexity. For transparency, I suggest the authors clearly specify a contact point through which data can be requested, and ideally clarify what conditions apply (institutional approval, deidentified data only, etc.).

If these minor revisions are implemented, I would be pleased to support the manuscript’s acceptance.

Sincerely,

7. PLOS authors have the option to publish the peer review history of their article (what does this mean? ). If published, this will include your full peer review and any attached files.

**Do you want your identity to be public for this peer review?** For information about this choice, including consent withdrawal, please see our Privacy Policy .

Reviewer #2: No

---

## [Author Response · Author response to Decision Letter 2]

24 Jul 2025

Manuscript title: Real-time and immediate effects of backward walking exercise on pain intensity and lumbopelvic movement control in individuals with chronic non-specific low back pain with lumbar flexion syndrome

Manuscript ID: PONE-D-24-59535R2

Reviewer #2:

1. Thank you for the opportunity to re-evaluate the revised manuscript. I appreciate the authors’ thoughtful and detailed responses to my previous comments, as well as the substantive revisions made throughout the manuscript. The additions – including clarification of the sample size calculation, expanded discussion of generalizability, revised figures, and improvements to terminology and grammar – have significantly strengthened the paper. Most of my concerns have been addressed appropriately.

Response: Thank you for your compliment and we are pleased to learn that our previous responses helped addressed most of the concerns raised in last round of review.

2. However, I would like to request a few minor revisions prior to recommending acceptance: The inclusion of ICCs for all muscle groups is appreciated, and the discussion of the factors influencing the low reliability of the right erector spinae is thorough. However, I suggest that the main text (particularly in the Results and Discussion) include explicit cautionary language wherever findings involving this muscle are interpreted, to ensure that readers are aware of the limitations in reliability.

Response: Thank you for your comments and recommendation. As advised, we have further revised our Results and Discussion by explicitly highlighting the precaution needed when interpreting the findings related to right erector spinae for its relatively low ICC as compared to the remaining list of muscles being investigated in this study. This now reads as, “The result indicated that all EMG measurements demonstrated good to excellent repeatability except right ES, and hence, interpretation of findings related to analysis of the right ES requires adequate caution for its observed variability of EMG magnitude.” (Lines 388-391, Results).

For Discussion section, it now reads as, “Although the level of reliability of right erector spinae EMG measurement is considered to be moderate with an ICC value of 0.558, it is indeed the lowest amongst the listed muscles. Therefore, interpretation of results involving EMG analysis of the right erector spinae would require extra caution irrespective to significant and insignificant findings statistically i.e., the significant increase in right ES:right GMax activity ratio during the 3rd minute of BW training and significant decrease in the same ratio found during the 12th minute of FW training, and those statistically insignificant comparisons reported in Tables 9, 12 and 13.” (Lines 626-632, Discussion)

3. While the discussion now notes the potential for Type I error and small effect sizes, I still recommend that the authors slightly temper the language used in key interpretive statements.

Response: Thank you for your comments and suggestion. Some refinement of the tone of our writing for the key interpretive statements was made with the aim to balancing the adequacy and accuracy of the discussion with respect to the potential Type I error and small effect size of our study (Lines 543, 547, 568, 603, 606, 650, 660-1, and 692).

4. The revised data availability statement notes that data are available upon request due to the dataset’s complexity. For transparency, I suggest the authors clearly specify a contact point through which data can be requested, and ideally clarify what conditions apply (institutional approval, deidentified data only, etc.).

Response: The data availability statement has been revised as, “Deidentified data are available upon request made to the corresponding author” (Entry at the online submission system).

5. If these minor revisions are implemented, I would be pleased to support the manuscript’s acceptance.

Response: We would like to thank the reviewer for the time and effort spent on reviewing our manuscript. We hope that we have adequately addressed the outstanding issues specified by the reviewer in this revised version. Thank you again for considering our revised manuscript.

End of the response letter

---

## [Editor Report · Decision Letter 2]

5 Aug 2025

Real-time and immediate effects of backward walking exercise on pain intensity and lumbopelvic movement control in individuals with chronic non-specific low back pain with lumbar flexion syndrome

PONE-D-24-59535R2

Dear Dr. Sharon MH Tsang,

We’re pleased to inform you that your manuscript has been judged scientifically suitable for publication and will be formally accepted for publication once it meets all outstanding technical requirements.

Kind regards,

Holakoo Mohsenifar

Academic Editor

PLOS ONE
---

## [Editor Report · Acceptance letter]

PONE-D-24-59535R2

PLOS ONE

Dear Dr. Tsang,

I'm pleased to inform you that your manuscript has been deemed suitable for publication in PLOS ONE. Congratulations! Your manuscript is now being handed over to our production team.

Kind regards,

on behalf of

Dr. Holakoo Mohsenifar

Academic Editor

PLOS ONE